# Matrix mechanics regulates epithelial defence against cancer by tuning dynamic localization of filamin

Shilpa P. Pothapragada[1], Praver Gupta [1], Soumi Mukherjee [1,2] & Tamal Das [1✉]

In epithelia, normal cells recognize and extrude out newly emerged transformed cells by competition. This process is the most fundamental epithelial defence against cancer, whose occasional failure promotes oncogenesis. However, little is known about what factors determine the success or failure of this defence. Here we report that mechanical stiffening of extracellular matrix attenuates the epithelial defence against HRas$^{V12}$-transformed cells. Using photoconversion labelling, protein tracking, and loss-of-function mutations, we attribute this attenuation to stiffening-induced perinuclear sequestration of a cytoskeletal protein, filamin. On soft matrix mimicking healthy epithelium, filamin exists as a dynamically single population, which moves to the normal cell-transformed cell interface to initiate the extrusion of transformed cells. However, on stiff matrix mimicking fibrotic epithelium, filamin redistributes into two dynamically distinct populations, including a new perinuclear pool that cannot move to the cell-cell interface. A matrix stiffness-dependent differential between filamin-Cdc42 and filamin-perinuclear cytoskeleton interaction controls this distinctive filamin localization and hence, determines the success or failure of epithelial defence on soft versus stiff matrix. Together, our study reveals how pathological matrix stiffening leads to a failed epithelial defence at the initial stage of oncogenesis.

[1] TIFR Centre for Interdisciplinary Sciences, Tata Institute of Fundamental Research Hyderabad (TIFR-H), Hyderabad 500 046, India. [2]Present address: Department of Biology, Purdue University, West Lafayette, IN 47907, USA. ✉email: tdas@tifrh.res.in

In the micro-ecosystem of epithelial tissues, epithelial cells display an extraordinary ability to maintain tissue homeostasis in the face of incessant sprouting of transformed cells[1–4]. In general, any newly emerged transformed cell gets actively extruded out of the tissue by neighbouring normal cells[2,3,5–8]. This process is the fundamental immune system-independent epithelial defence against cancer (EDAC)[1,2,4,9–13] and belongs to a larger class of tissue quality-control processes, collectively known as cell competition[1,3]. Cell competition, in general, describes any process involving a struggle for space between two cell populations, wherein the 'winner' population eliminates the 'loser' population in a non-cell-autonomous manner. It plays critical roles in *Drosophila* wing epithelial development[2,3,14], mammalian tissue dynamics[15], skin development[16], and tumour suppression in thymus[17]. EDAC, on the other hand, refers specifically to the elimination of the transformed cells that either lack a tumour suppressor protein[18] or express a constitutively active form of an oncogene such as HRAS[1,9,12,19]. Oncogene-expressing transformed cells leave the epithelial layer in the form of apical extrusion or basal delamination only if these cells are surrounded by normal epithelial cells[9]. Importantly, occasional failure of EDAC leads to hyperplastic proliferation and cancer[1,2]. Despite some recent progress towards uncovering the molecular players of EDAC[9,12,19,20], key biochemical and biophysical factors which determine whether newly emerged transformed cells get extruded from the tissue or can continue to belong there remain largely elusive[7].

Nevertheless, it is evident that EDAC-mediated removal of transformed cells requires extensive reorganization of force-bearing cytoskeletal elements in surrounding normal cells, particularly at the interface between normal and transformed cells[5,19,21–23]. These observations indicate possible mechanical modulations of EDAC[5,6,9,21–25]. One would then presume that EDAC might respond to the mechanical properties of tissue microenvironment, including the extracellular matrix (ECM) stiffness, and this parameter could be a critical factor in determining the success or failure of EDAC. In fact, matrix stiffness plays a critical role in cancer progression and metastasis at the advanced oncogenesis stage[26–29]. At this stage, cancer-associated stiffening of ECM propels transformed cells to disrupt the mono-layered architecture of epithelium, proliferate without contact inhibition, and migrate out of primary tumour[26,28]. In contrast, the role of matrix stiffness on the initial pre-malignant stage of carcinogenesis, including EDAC, remains mostly unknown. Such lack of knowledge is surprising given that one should expect the effect of matrix stiffness to be very pronounced during the initial transformation events wherein transformed cells still bear the modulations imposed by the host tissue microenvironment. Interestingly, pathological conditions like fibrosis, hyperactive wound healing, obesity, and ageing can induce tissue stiffening, and all of these conditions correlate with high cancer risk[30,31]. However, it remains elusive whether such pre-existing fibrotic or pathologically stiffened tissue affects the success or failure of EDAC and if it indeed does, what could be the connecting molecular mechanisms.

Here, we discover matrix mechanics as a key microenvironmental factor that decides the success or failure of EDAC. We further elucidate the molecular mechanism underlying it, specifically demonstrating that matrix stiffness-dependent intracellular localization of filamin is key to the regulation of EDAC.

## Results

### Matrix stiffness regulates the extrusion of HRas[V12]-transformed cells in EDAC.
First, to examine whether matrix stiffness could really alter the outcome of EDAC, we used a well-established mammalian model of EDAC[9,13,32] (Fig. 1a, b) and performed EDAC experiments on ECM of varying stiffness that satisfied the range of physiological and pathological elasticity of healthy and fibrotic epithelial matrix[30,33] (Fig. 1c). The EDAC model involved normal epithelial cells competing against and eliminating transformed cells expressing a constitutively active HRas protein (HRas[V12]) (Fig. 1a, b). A tetracycline-inducible promoter controlled HRas[V12] expression, which enabled us to initiate the competition process when intended. We first mixed normal or wild-type epithelial cells (MDCK-WT) and cells with tetracycline-inducible GFP-tagged HRas[V12] stably integrated into the genome (MDCK-GFP-HRas[V12]) in 40:1 ratio (Supplementary Fig. 1a) and cultured a mosaic monolayer of these populations in the absence of tetracycline. Subsequently, the addition of a stable tetracycline-derivative, doxycycline, in the medium triggered HRas[V12] expression, which became apparent at 30 min post-induction. HRas[V12]-transformed cells started rounding up after 3 h, and most of them extruded within eight-to-ten hours (Supplementary Fig. 1b, Supplementary Video 1, Fig. 1b). We performed this experiment on collagen I-coated hydrogel substrates of six individual discrete stiffness values, having an elastic modulus of 1.2, 4, 11, 23, 35, or 90 kPa[30,33] (Fig. 1c). For each stiffness, we counted the fraction of HRas[V12]-expressing colonies that extruded at 6 h post-induction (Fig. 1c) and 4 h post-induction (Supplementary Fig. 1c) and observed that this fraction decreased drastically on substrates stiffer than 11 kPa (Fig. 1c). Relevantly, this result does not depend on the ECM protein coating since we also observed a similar inhibitory effect of ECM stiffness on EDAC when we performed the experiments on laminin and basement membrane extract (BME)-coated substrates (Supplementary Fig. 1d). We thus classified 1.2, 4, and 11 kPa substrates as 'soft', mimicking healthy epithelium, and 23, 35, and 90 kPa substrates as 'stiff' mimicking fibrotic epithelium (Fig. 1c). Interestingly, previous studies had recorded an increase in ECM stiffness from 0.1 to 5 kPa in healthy epithelium to 25–100 kPa in fibrotic epithelium[34,35], which justified our classification and the stiffness transition value (>11 kPa) that we found. We further found that mutant cell extrusion in absence of doxycycline was significantly low on soft substrates (Supplementary Fig. 1e) when compared with doxycycline-induced extrusion, confirming the active extrusion of HRas[V12]-expressing cells on soft substrate. We next wondered what happened to the transformed cells that did not extrude on the stiff substrate. These cells remained in the monolayer and eventually showed long basal protrusions and prominent basal actin fibres (Fig. 1d). These features were absent in normal cells. On observing these cells up to 60 h post-induction, we noticed that HRas[V12]-transformed cells remained in the monolayer, started dividing, and the colony size expanded (Supplementary Fig. 1f, Supplementary Video 2). To further check if the effect of ECM stiffness on EDAC is not cell-line specific, we performed cell competition experiments between normal and HRas[V12]-expressing cells in two other epithelial cell lines, namely Eph4 and Caco-2, by growing them either on 4 kPa (soft) or on 90 kPa (stiff) ECM. In both cell lines, we observed a significantly less fraction of extruded colonies of HRas[V12]-expressing cells on 90 kPa ECM than on 4 kPa ECM (Supplementary Fig. 1g), consolidating the generality of our observations with MDCK cells. Collectively, these results demonstrate that ECM stiffness has a decisive effect on the efficacy of EDAC-associated cell competition, where stiff ECM inhibits the elimination of transformed cells.

### Differential localization of filamin on soft versus stiff matrix determines EDAC efficacy.
We next looked for the molecular mechanism by which stiff ECM inhibited EDAC-induced cell

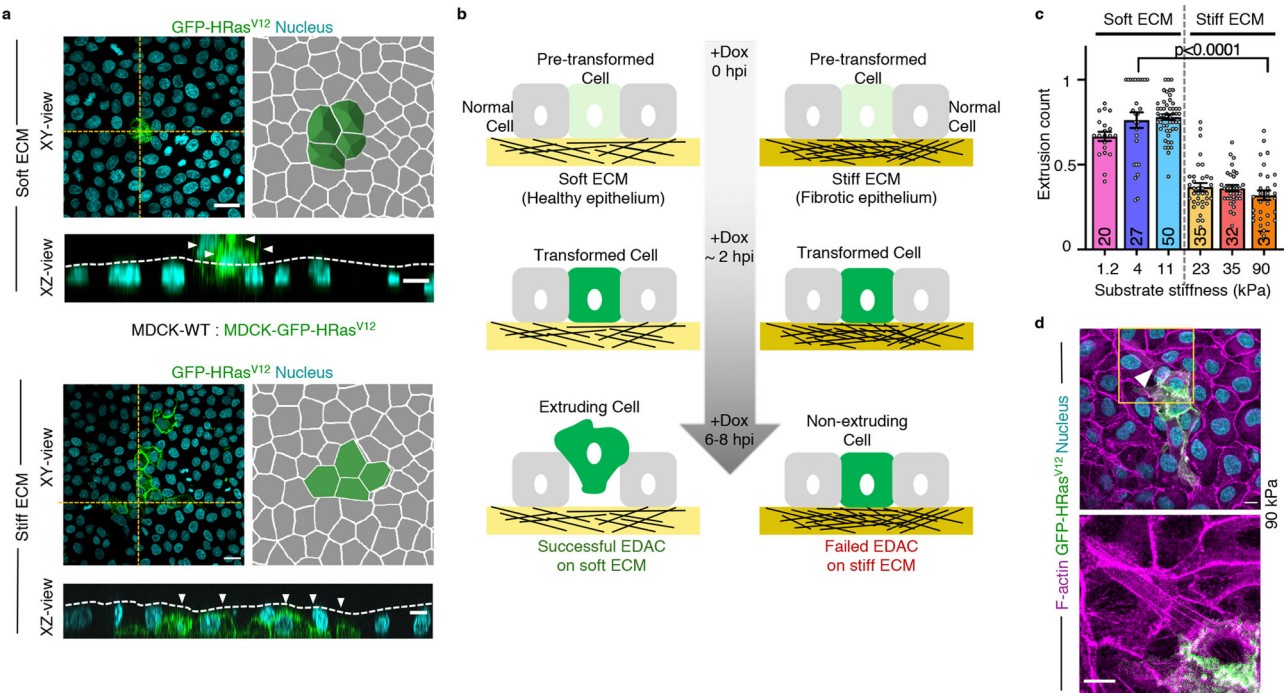

**Fig. 1 Matrix stiffening attenuates extrusion of HRas[V12]-transformed cells during EDAC. a** Fluorescence images of GFP-HRas[V12] expressing colony extrusion on soft (4 kPa) and stiff (90 kPa) substrates in *XY*-plane; followed by an illustration depicting the visual metric employed to quantify extrusion. Rounded-up cells expressing GFP-HRas[V12] (as seen on soft ECM) are taken as extruded. Non-extruded GFP-HRas[V12] cells remain in-plane with other cells, as evident on stiff ECM. (*Bottom panels*) The yellow-dotted lines visually guide the epithelium surface in *XZ*-plane. GFP-HRas[V12] cells extruded over this surface on soft ECM (*top*) whereas they remained within this surface on stiff ECM (*bottom*). White arrowheads indicate GFP-HRas[V12] cells. **b** Diagram representing different phases of extrusion of transformed cells and stiffness-dependent outcome of EDAC. **c** Scatter bar plot depicting the fraction of GFP-HRas[V12] expressing colonies extruded over substrates of varying stiffness at 6 hpi. Distinct decrease in extrusion of transformed cells observed with increase in substrate stiffness. The number of colonies counted is indicated inside each bar. Data are mean ± s.e.m. collected over three independent biological replicates. Statistical significance was assessed using Mann–Whitney *t*-test (two-tailed). $p = 2.8490e{-}08$. **d** Cytoskeletal morphology of non-extruded colonies over stiff ECM at 24 hpi. White arrowheads indicate basal actin fibres associated with HRas[V12]- cells on stiff ECM (90 kPa), stained with AlexaFluor647-Phalloidin. Inset: Magnified view of the yellow-boxed region with actin fibres pointed out by white arrowheads. Scale bars = 20 μm (**a**, XY-view), 10 μm (**a**, XZ-view; **d**).

extrusion. Extrusion of transformed cells requires remodelling of the actin cytoskeleton in the normal cells that directly interface with the former[19,21–23]. Since ECM stiffness alters the cellular localization of many force-sensitive cytoskeleton-related proteins[36,37,38], we initially checked whether any actin-binding or actin-crosslinking proteins showed a localization difference on soft versus stiff ECM in normal cells (Supplementary Fig. 2). An actin filament cross-linking protein, FilaminA (FLNA, referred to as filamin hereafter), emerged as the most promising candidate (Fig. 2a), considering its stiffness-sensitive perinuclear localization in normal cells (Fig. 2a, b, Supplementary Fig. 3a). On soft ECM, filamin localized to cytoplasm and cell–cell interface, while on stiff ECM, a significant fraction of filamin molecules localized to perinuclear region (Fig. 2a, b, Supplementary Fig. 3a). Actin counter-staining and subsequent confocal microscopy (Supplementary Fig. 3b) and ultrastructure expansion microscopy (U-ExM)[39] (Fig. 2c, Supplementary Fig. 3c) revealed that perinuclear filamin molecules co-localized with perinuclear actin cytoskeleton on the stiff substrate. While previous work had reported filamin accumulation at normal cell-transformed cell interface and depletion of filamin in normal cells abrogated EDAC[19], this stiffness-sensitive perinuclear localization is a unique finding. Relevantly, on stiff ECM, filamin showed depleted interfacial localization and enhanced perinuclear localization than on soft substrate (Fig. 2b). We next asked whether this filamin localization pattern could be relevant to EDAC. In the normal cells interfacing with HRas[V12]-expressing cells, filamin showed

increased interfacial fraction (Fig. 2b), indicating that during competition, filamin relocates to cell–cell interface[19]. We, therefore, hypothesize that perinuclear filamin on stiff ECM perhaps sequesters this interfacial pool, making a large fraction of filamin unavailable for EDAC. To test this hypothesis, we stably overexpressed filamin in normal cells to compensate for the loss of interfacial fraction on stiff ECM (Fig. 2d, *left*). The filamin overexpressing cells showed clear interfacial and perinuclear filamin pools, at the same time, on stiff ECM (Fig. 2d, *left*, Supplementary Fig. 3d). We then quantified the fraction of extruded HRas[V12]-expressing colonies during the competition between filamin-overexpressing normal cells and HRas[V12]-expressing cells (Fig. 2d, *right*). Filamin overexpression indeed rescued the extrusion of transformed colonies on stiff ECM, rendering EDAC insensitive to substrate stiffness (Fig. 2d). Finally, we validated that the differential localization of filamin on soft versus stiff ECM is not matrix coating-specific by exploring filamin localization in MDCK cells, grown on soft and stiff ECM coated with either laminin or BME (Supplementary Fig. 3e). We also validated that the same is not cell line-specific by reproducing the stiffness-dependent filamin localization in Eph4 and Caco-2 cells (Supplementary Fig. 3f). These results, together, showed that differential localization of filamin on soft versus stiff ECM plays a critical role in EDAC.

We then performed direct experiments to study the dynamics of filamin localization during EDAC on soft versus stiff ECM and to elucidate the effect of perinuclear sequestration of filamin on

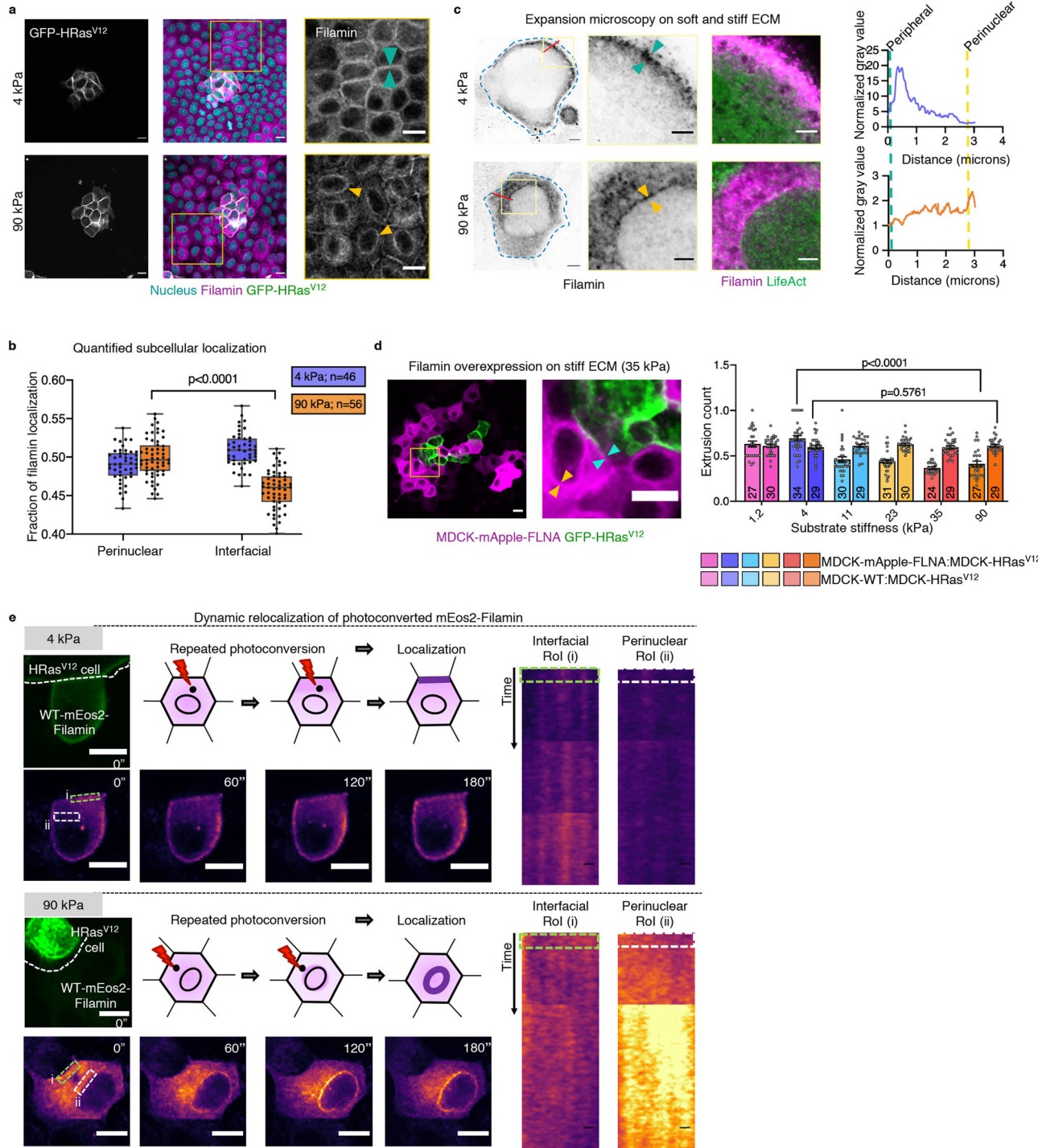

stiff ECM during EDAC (Fig. 2e, Supplementary Videos 3 and 4). To this end, we expressed moderate levels of filamin tagged with a green-to-red photoconvertible fluorescent protein, mEos2, in normal cells and selected those mEos2-filamin expressing cells that interfaced with at least one HRas[V12]-expressing cell. We then photoconverted a population of mEos2-tagged filamin molecules from green to red, at a point nearly halfway between the cell–cell interface and the cell nucleus (Fig. 2e). We subsequently studied where those photo-converted filamin molecules localized during EDAC. On repeated cycles of photoconversion, filamin molecules invariably moved to the cell–cell interface on soft ECM (Fig. 2e, *top panels*, Supplementary Video 3). On stiff ECM, however, photoconverted filamin predominantly moved to the perinuclear

region (Fig. 2e, *bottom panels*, Supplementary Video 4). Moreover, a separate set of photobleaching experiments in mApple-filamin expressing cells showed that on stiff ECM, filamin had two dynamically different populations—interfacial and perinuclear (Supplementary Fig. 3g, h), in terms of the speed of recovery after photobleaching. On stiff ECM, the perinuclear population was more dynamic and recovered faster than the interfacial population (Supplementary Fig. 3g, h). A comparison of the filamin population dynamics between soft and stiff ECM interestingly revealed that the perinuclear pool on stiff ECM matched very closely to that of the interfacial population on soft ECM (Supplementary Fig. 3h). This indicated that from a dynamics perspective, the perinuclear population of filamin on

**Fig. 2 ECM stiffness-dependent localization of filamin determines EDAC efficacy. a** Immunostaining image of mosaic monolayer of MDCK-WT:MDCK-GFP-HRas[V12] cultured on soft (*top panels*) and stiff (*bottom panels*) ECM. From *left* to *right*: GFP-HRas[V12], filamin in the mosaic monolayer and magnified view of the boxed regions. Cyan arrowheads indicate interfacial filamin enrichment on soft ECM, and yellow arrowheads indicate perinuclear filamin localization on stiff ECM. Scale bars = 10 μm. **b** Box-and-whiskers plot depicting the fraction of filamin mean fluorescence intensity at perinuclear and interfacial regions, per cell, on soft (4 kPa) and stiff (90 kPa) ECM. Statistical significance was assessed using an unpaired Student *t*-test with Welch's correction (two-tailed). $p = 6.35542e{-}09$. For boxplots, centre line denotes median, box displays the interquartile range, whiskers indicate range not including outliers (1.5× interquartile range). **c** Post expansion images of LifeAct-GFP MDCK cells stained for endogenous filamin and cultured on either 4 or 90 kPa PAA gels. Filamin localization differences are indicated by yellow arrowhead (perinuclear) and cyan (interfacial) in the insets. Scale bar = 2 μm; Inset = 1 μm. Fluorescence intensity line scan plots for the red line marked in the filamin channel. **d** (*Left*) Fluorescence images of a mosaic monolayer of filamin-over expressing MDCK cells and MDCK-GFP-HRas[V12] co-cultured on a stiff substrate with a magnified view of the boxed region. Yellow and cyan arrowheads indicate perinuclear and interfacial filamin accumulation respectively. (*Right*) Scatter bar plot depicting the fraction of GFP-HRas[V12] expressing colonies extruded over substrates of varying stiffness. For each stiffness, left bars are for mock MDCK-WT:MDCK-GFP-HRas[V12] and right bars are for MDCK-mApple-FLNA:MDCK-GFP-HRas[V12] mosaic populations. Stable over-expression of filamin in surrounding cells rescued extrusion of transformed populations on the stiff substrate. The number of colonies counted is indicated inside each bar. Data are mean ± s.e.m. collected over three independent biological replicates. Statistical significance was assessed using unpaired Student's *t*-test with Welch's correction (two-tailed). Significance value (when $p < 0.0001$) $p = 9.27612e{-}09$. Source data (**b, d**) are provided as a Source Data file. **e** Photoconversion of mEos2-filamin to study filamin localization dynamics. mEos2-filamin cell interfacing with an HRas[V12] cell was stimulated and tracked for 180 s, as depicted in the schematic. Snapshots of mEos2-filamin dynamics indicate dynamic changes in localization. Kymographs of interfacial (i, green box) or perinuclear (ii, white box) regions show enrichment of filamin at the interface (soft ECM, *top panel*) or perinuclear region (stiff ECM, *bottom panel*). Scale bars = 5 μm.

stiff ECM behaved more similarly to the interfacial population of filamin on soft ECM than to the interfacial population on stiff ECM (Supplementary Fig. 3h). Importantly, photobleaching experiments did not reveal any dynamic differentiation of filamin localization on soft ECM where a prominent perinuclear population was anyway missing. Both fixed-cell and dynamic experiments provided converging evidence proving that the perinuclear cytoskeleton acted like a sink on stiff ECM, by reducing the fraction of filamin molecules available for EDAC at the interface between normal and transformed cells. Therefore, on stiff ECM, the interaction between normal and transformed cells fails to initiate the extrusion of transformed cells.

**Cdc42 and perinuclear cytoskeleton determine differential filamin localization**. We next asked what molecular signalling pathways decided the differential filamin localization on soft and stiff ECM. To this end, small RhoGTPase Cdc42 is one of the strongest filamin-binding proteins with a very high interaction score of 0.979 in STRING protein interaction database (https://string-db.org/). Given that RhoGTPases, in general, play an important role in mechanotransduction, we speculated whether Cdc42 might have different activation on soft versus stiff ECM. Transfecting the normal cells with a förster resonance energy transfer (FRET)-based Cdc42 activity sensor[40] indeed revealed stiffness-dependent differences in Cdc42 activity (Fig. 3a). Cdc42 activity at the cell–cell interface was broader and stronger (Fig. 3b) on soft ECM than on stiff ECM. As an alternative representation for Cdc42 activity, staining for a Cdc42-activating guanine nucleotide exchange factor (GEF), Tuba[41], also indicated higher interfacial Cdc42 activation (Fig. 3c). Localization of Tuba to the cell–cell interface was clearly more prominent on soft ECM than on stiff ECM (Fig. 3c). We then asked whether the interfacial localization of filamin on soft ECM might be a consequence of interfacial activation of Cdc42. To this end, we treated the normal cells cultured on soft ECM with a Cdc42-activity inhibitor, ML141, and studied the localization of filamin upon this inhibitor treatment (Fig. 3d). This experiment revealed that upon ML141 treatment, the interfacial localization of filamin vanished while a faint perinuclear filamin ring appeared even on soft ECM (Fig. 3d, e. Supplementary Fig. 4a). Interestingly, we could also induce interfacial localization on stiff ECM by a reverse manipulation, where we expressed a constitutively active Cdc42[Q61L] in some cells (Fig. 3f, Supplementary Fig. 4b). As compared to

surrounding non-transfected cells, Cdc42[Q61L]-expressing cells showed enhanced interfacial localization of filamin, on both soft and stiff ECM. Also, the stiff ECM-specific perinuclear filamin ring disappeared in Cdc42[Q61L]-expressing cells (Fig. 3f, *left*). In contrast, Cdc42[Q61L] itself showed prominent interfacial as well as perinuclear localization on stiff ECM (Fig. 3f, *middle*). Hence, taken together, these experiments proved that Cdc42 activation drives the interfacial localization of filamin, especially on soft ECM. However, given that filamin and Cdc42[Q61L] did not co-localize at the perinuclear region (Fig. 3f, *right*), they indicated that Cdc42 might not be directly responsible for the perinuclear localization of filamin on stiff ECM. While these experiments depicted cell-autonomous interaction between Cdc42 and filamin, cell competition is a non-cell-autonomous event. Considering that filamin accumulated at the interface between normal and transformed cells during competition, we asked whether Tuba might also show similar accumulation. Tuba membrane localization was indeed promoted at the interface between normal and HRas[V12]-transformed cells, where filamin accumulated (Supplementary Fig. 4c).

We then asked what recruits filamin to the perinuclear cytoskeleton on stiff ECM. In non-epithelial cells, a refilin family protein, FAM101B or refilinB, stabilizes perinuclear actin networks by associating with filamin[42] (Fig. 4a, Supplementary Fig. 4d). Thus, FAM101B seemed to be a likely candidate that recruited filamin to perinuclear cytoskeleton on stiff ECM. As preliminary evidence, we indeed found that on soft ECM, FAM101B distributed all over the cytoplasm (Supplementary Fig. 4e). However, on stiff ECM, it co-localized with filamin predominantly in the perinuclear region (Supplementary Fig. 4e). To test whether filamin–FAM101B interaction might be responsible for the perinuclear localization of filamin, we generated mutant filamin, FLNA-[19-22] or dnFLNA, that carried only four filamin repeats. dnFLNA had been known to have a dominant-negative effect on the interaction between endogenous filamin and FAM101B[42]. Stably expressing dnFLNA in normal cells indeed decreased the perinuclear localization of endogenous filamin and increased its interfacial pool (Fig. 4b, f). We moreover generated a dominant-negative FAM101B (dnFAM101B) mutant[42] that lacked one of the filamin-binding domains, BD2. Stable expression of this mutant also decreased the perinuclear localization of filamin and increased its interfacial pool (Fig. 4c, f). Together, these experiments indicated that filamin–FAM101B interaction plays a crucial role in the localization of filamin to the

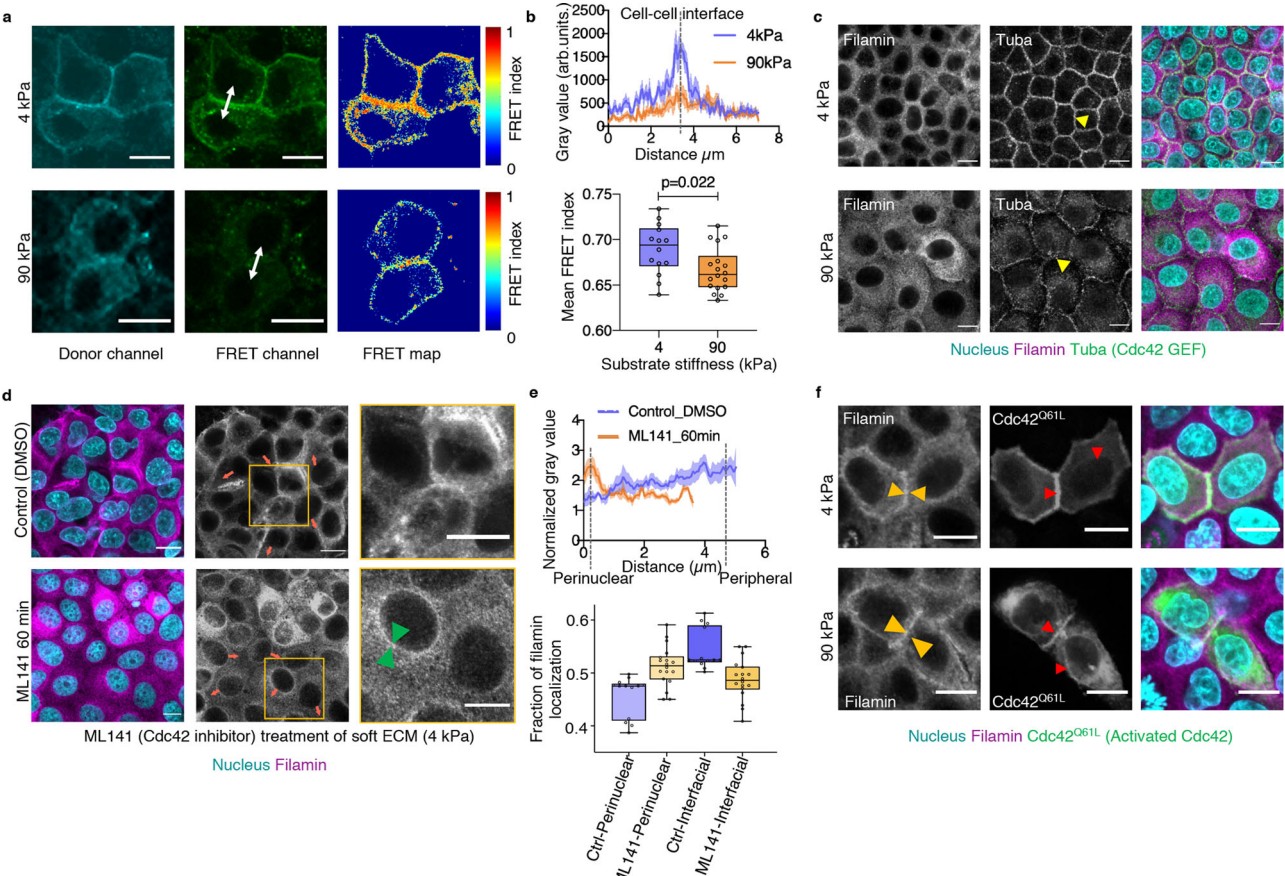

**Fig. 3 Interaction with Cdc42 determines interfacial filamin localization. a** Raichu-Cdc42 FRET biosensor expressed in MDCK-WT cells cultured on soft (*top*) and stiff (*bottom*) ECM. **b** (*Top*) Line scan of grey values from the FRET channel (representative white double-arrowed line in (**a**) from multiple cells cultured on both soft and stiff ECM indicate higher FRET values at cell–cell interface on soft ECM. Bold line plot depicts the mean grey values with the shaded region accounting for mean ± s.e.m of grey values from $n = 10$ (4 kPa) and 8 (90 kPa) cells. (*Bottom*) Box-and-whiskers plot of mean FRET index from soft and stiff ECM shows a significant reduction in cells cultured on stiff ECM, indicative of higher Cdc42 activity on soft ECM. Statistical significance was assessed using unpaired Student's *t*-test with Welch's correction (two-tailed). $n = 14$ (4 kPa) and 18 (90 kPa). **c** Immunofluorescence images of MDCK-WT cells co-stained for filamin and Tuba (Cdc42-GEF). Yellow arrowhead indicates enrichment of Cdc42-GEF at the interfacial region on soft ECM (top) and at the perinuclear region on stiff ECM (bottom). **d** Immunofluorescence images of MDCK-WT cells cultured on soft ECM treated with Cdc42 inhibitor: control DMSO (*top*) or ML141 (*bottom*) and immunostained for filamin. Green arrowhead indicates enrichment of filamin at the perinuclear region on soft ECM (*bottom*) post-treatment with ML141. Red arrowheads indicate the direction of line scans, starting from near the nucleus and extending to periphery **e.** (*Top*) Line scan of normalized grey values of filamin along red lines from multiple cells in (**d**) shows perinuclear filamin enrichment peaks with ML141 treatment (*orange line*). The bold line plot depicts the mean grey values with the shaded region accounting for mean ± s.e.m of grey values from $n = 5$ cells. (*Bottom*) Box-and-whiskers plot of the fraction of filamin localization in cells cultured on soft ECM and treated with ML141 (*orange*) or without treatment (*Ctrl, blue*). $n = 13$ (Ctrl) and 18 (ML141 treated) cells. **f** MDCK-WT cells transfected with constitutively active Cdc42 (Cdc42$^{Q61L}$) and immunostained for filamin. Yellow arrowheads indicate increased interfacial enrichment of filamin on stiff ECM. Red arrowheads indicate enriched areas of constitutively active Cdc42. Scale bars = 10 µm. For boxplots, the centre line denotes the median, the box displays the interquartile range, whiskers indicate the range not including outliers (1.5× interquartile range). Source data (**b**, **e**) are provided as a Source Data file.

perinuclear actin cytoskeleton. Interestingly, the perinuclear actin cytoskeleton is connected to the nuclear lamina via the LINC (linker of nucleoskeleton and cytoskeleton) complex. This linkage enables direct transmission of extracellular cues such as matrix mechanics to the nuclear force-sensing machinery[43–45] (Fig. 4a). In fact, using a FRET-based molecular tension sensor module[46], inserted in the middle of a LINC complex protein, Nesprin1, we measured lower FRET efficiency (Supplementary Fig. 4f). This result implied higher LINC complex tension on stiff ECM than on soft ECM (Supplementary Fig. 4f). We, therefore, asked whether cytoskeleton–nucleoskeleton mechanical linkage could additionally be a critical factor in the stiffness-sensitive perinuclear localization of filamin. To this end, we first delinked the perinuclear cytoskeleton from nuclear lamina by disrupting the LINC complex with a dominant-negative Nesprin1 (dnNesprin1

or Nesprin1-KASH) lacking the cytoskeleton binding domain[44]. In another set of experiments, we disrupted the nucleoskeleton with a LaminB1 mutant (dnLaminB1or XLaminB1Δ2+)[47] that disassembled the nuclear lamina. Consequently, normal cells stably expressing dnNesprin1 or dnLaminB1 (Fig. 4d, e) showed decreased perinuclear localization of endogenous filamin (Fig. 4f), indicating that in addition to and perhaps upstream of FAM101B, the perinuclear cytoskeleton–nucleoskeleton mechanical linkage was indeed a critical factor for the perinuclear localization of filamin.

Taken altogether, these experiments involving manipulation of Cdc42 activity vis-a-vis inhibition of FAM101B–filamin interaction or disruption of LINC complex provided the molecular basis of the interfacial and perinuclear localization of filamin. They further implied that there could be a delicate competition

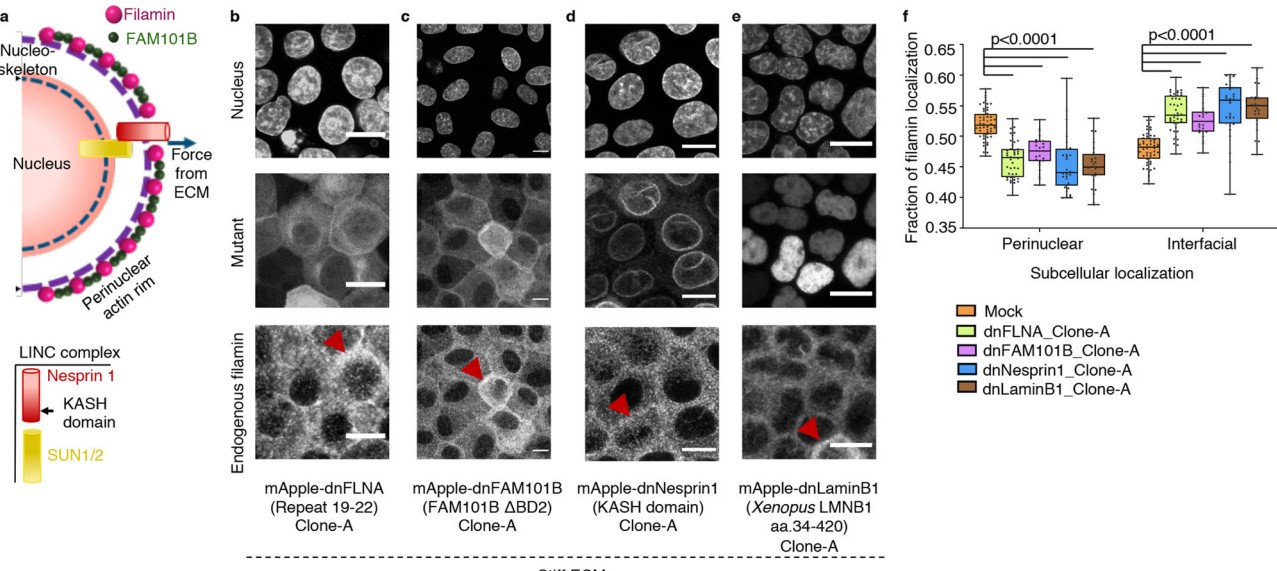

**Fig. 4 Interaction with perinuclear cytoskeleton determines perinuclear filamin localization. a** Schematic representation for filamin structure, depicting its interactions with FAM101B and F-actin. The filamin-FAM101B-actin localization at perinuclear space is enabled by the complex's interaction with LINC complex, which responds to extracellular force cues. **b–e** Immunofluorescence images of mApple-dnFLNA-MDCK (**b**), MDCK-mApple-dnFAM101B (**c**), mApple-dnNesprin1-MDCK (**d**) and mApple-dnLaminB1-MDCK (**e**) stained for filamin. Red arrowheads indicate interfacial filamin localization on stiff ECM. Scale bars = 10 μm. **f** Box-and-whiskers plot depicting the fraction of filamin mean fluorescence intensity at perinuclear and interfacial regions, per cell, on stiff (90 kPa) ECM; $n$ = 56 (mock), 43 (dnFLNA), 26 (dnFAM101B), 30 (dnNesprin1) and 24 (dnLaminB1) cells over three independent experiments. Statistical significance was assessed using an unpaired Student's $t$-test with Welch's correction (two-tailed). Significance values: $p$ = 2.56898e−16 (mock-dnFLNA), 1.13087e−09 (mock-dnFAM101B), 6.35474e−09 mock-dnNesprin1), 6.62199e−10 (mock-dnLaminB1). For boxplots, the centre line denotes the median, the box displays the interquartile range, whiskers indicate the range not including outliers (1.5× interquartile range). Source data (**b–e**) are provided as a Source Data file.

between filamin–Cdc42 interaction and filamin–perinuclear cytoskeleton interaction (Supplementary Fig. 4g). Filamin–Cdc42 interaction drives filamin molecules towards the cell–cell interface and is stronger on soft ECM than on stiff ECM (Supplementary Fig. 4g). In contrast, filamin–perinuclear cytoskeleton interaction drives them towards the perinuclear region and is stronger on stiff ECM than on soft ECM (Supplementary Fig. 4g).

**Rescuing EDAC on stiff ECM.** Having identified the molecular signalling pathway that favours filamin localization to perinuclear cytoskeleton on stiff ECM, we finally wondered whether perturbing this localization would restore EDAC on stiff ECM. To this end, we generated modified normal cells stably expressing the mutants that abolished perinuclear localization of filamin and increased interfacial filamin on stiff ECM (Fig. 4b–e, Supplementary Fig. 5a), including dnFLNA, dnFAM101B, dnNesprin1 or dnLaminB1. We created three clones per mutant to eliminate any clone-specific bias (Clone A: Fig. 4b–e; Clones B and C: Supplementary Fig. 5f–i). We then tested whether these cells with increased interfacial filamin could extrude the transformed cells on stiff ECM when the former surrounded the latter (Fig. 5a). Under this experimental condition, we indeed observed significantly increased extrusion of transformed on stiff ECM (Fig. 5b–e, Supplementary Figs. 5j–m). For example, stable dnFLNA expression in normal cells surrounding the HRas$^{V12}$-expressing colonies indeed rescued the extrusion of HRas$^{V12}$-transformed colonies on stiff ECM (Fig. 5b, Supplementary Fig. 5j). Similarly, normal cells stably expressing dnFAM101B could extrude HRas$^{V12}$-expressing colonies with equal efficacy, irrespective of substrate stiffness (Fig. 5c, Supplementary Fig. 5k). Finally, stably expressing either dnNesprin1 or dnLaminB1 in normal cells also rescued the extrusion of transformed cells on stiff substrate (Fig. 5d, e; Supplementary Fig. 5l, m, respectively).

Hence, all four mutants that reduced the perinuclear localization of filamin (Fig. 4b–e), either by disrupting filamin–FAM101B interaction (dnFLNA and dnFAM101B) or by disrupting the nuclear mechanotransduction (dnNesprin1 and dnLaminB1), also made cell competition more or less insensitive to ECM stiffness (Fig. 5b–e). Collectively, these experiments established the perinuclear localization of filamin on stiff ECM to be a clear molecular cause behind the failure of EDAC on stiff ECM and suggested possible therapeutic targets in future. Consolidating our experimental results, we discovered and established matrix mechanics as a crucial micro-environmental factor that decided the success or failure of EDAC and also elucidated the underlying molecular mechanism by which matrix mechanics regulated EDAC, thus integrating processes occurring across several length-scales (Fig. 5f).

## Discussion

Over the years, various studies have either implied or demonstrated that mechanical forces could be a crucial factor in all stages of EDAC and cell competition, including cell–cell sensing, cellular reorganization, and cell extrusion[1,5,6,21,24,25]. These studies have further shown that given a specific tissue- and mutation context, mechanical forces can have either indirect or direct effects on cell competition[5]. For example, forces can indirectly influence the elimination of transformed cells by modulating cell shape and the geometry of cell–cell interface[21]. Alternatively, they can directly induce transformed cell extrusion by compressing the 'loser' cells and triggering their apoptosis[24]. Nevertheless, given that cell–cell and cell–matrix forces are tightly regulated in epithelial tissue, the role of mechanical forces on EDAC could be a simple extrapolation of epithelial homoeostasis of cell density and the constraints imposed by epithelial architecture[29,48]. To this end, while previous works have elucidated a dynamic modulation

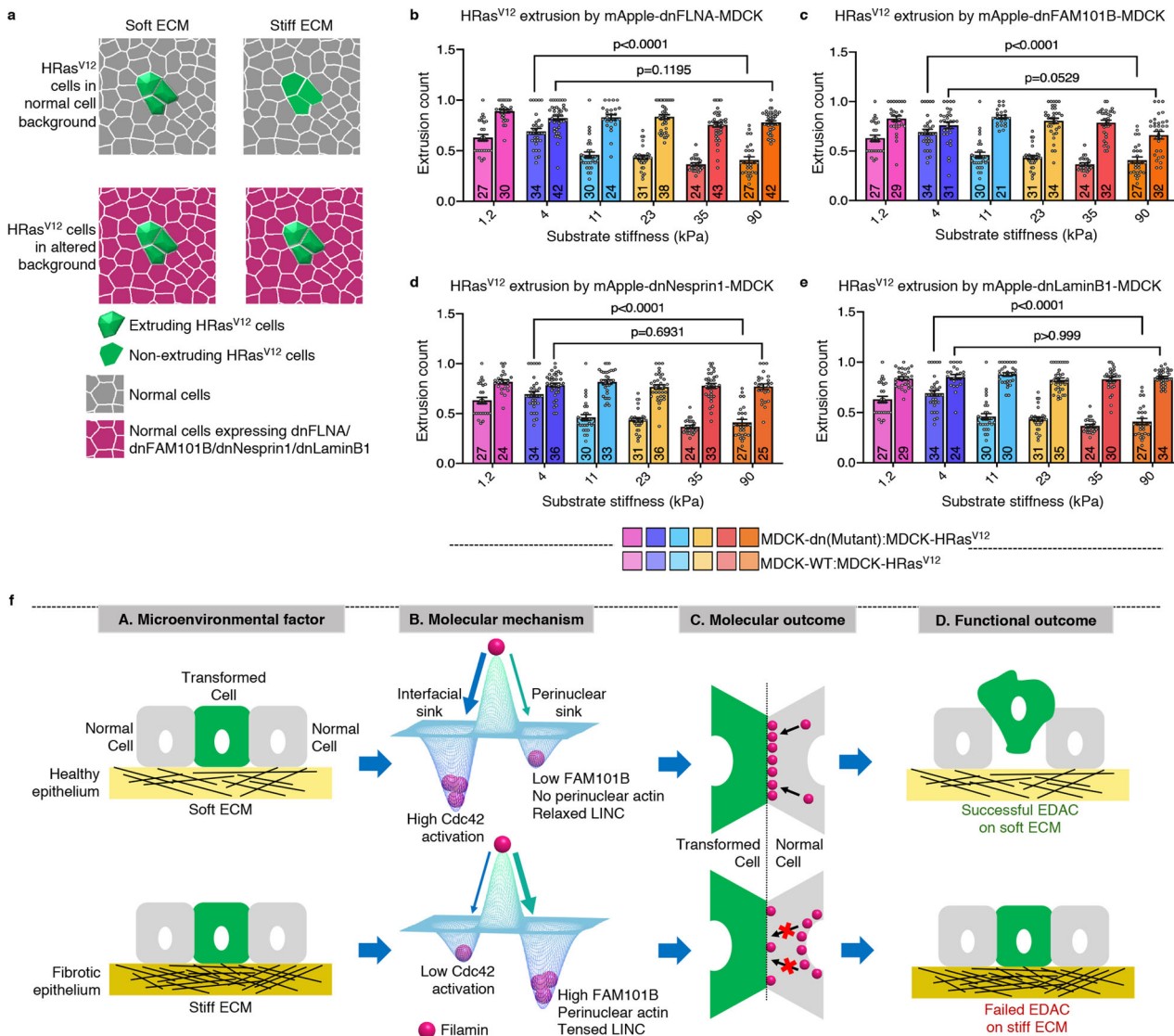

**Fig. 5 Rescuing EDAC on stiff ECM. a** An illustration showing the experimental design for rescuing EDAC on stiff ECM and the effect of different mosaic populations on extrusion of transformed cells on stiff ECM. **b–e** Scatter bar plots depicting the fraction of GFP-HRas$^{V12}$ expressing colonies extruded over substrates of varying stiffness. For each stiffness, left bars are for mock MDCK-WT: MDCK-HRas$^{V12}$GFP and right bars are for mApple-dnFLNA-MDCK:MDCK-GFP-HRas$^{V12}$ (**b**), MDCK-mApple-dnFAM101B:MDCK-GFP-HRas$^{V12}$ (**c**), mApple-dnNesprin1-MDCK:MDCK-GFP-HRas$^{V12}$ (**d**) or mApple-dnLaminB1-MDCK:MDCK-GFP-HRas$^{V12}$ (**e**) mosaic populations. Stable expression of dnFLNA, dnFAM101B, dnNesprin1 or dnLaminB1 in surrounding cells rescued extrusion of transformed populations on the stiff substrate. The number of colonies counted is indicated inside each bar. Data are mean ± s.e.m. collected over three independent biological replicates. Statistical significance was assessed using Unpaired *t*-test with Welch's correction (two-tailed). Significance value (when *p* < 0.0001) *p* = 9.27612e−09. Source data (**b–e**) are provided as a Source Data file. **f** Summary of the integrative mechanism that connects the processes occurring at different length scales. We elucidate matrix mechanics, a microenvironmental factor (**a**), influences the molecular dynamics of filamin (**b** and **c**) to determine the outcome of EDAC (**d**).

of cell–cell junctions during cell competition[21–23], the precise role of cell–matrix interaction has remained elusive. Specifically, what has remained mostly speculative is the role of tumour microenvironment[1] and the consequent mechanical cues on the strength and outcome of EDAC. To this end, while looking for the microenvironmental factors that might decide the success or failure of EDAC, our study reveals that an abnormally stiffened ECM prevents EDAC-mediated elimination of HRas$^{V12}$-transformed cells from an epithelial monolayer (Fig. 1a). ECM stiffness, thus, emerges to be a critical mechanical parameter of the tumour microenvironment regulating the basic immune system-independent EDAC (Fig. 5f).

Importantly, over the last two decades, it has also emerged that mechanical cues originating from the ECM play a decisive role in

cellular fate, form, and function during stem cell differentiation and organogenesis[27,33,37,45,49,50]. We also know that the synergistic integration of biochemical and mechanical signalling influenced by tissue stiffening supports the metastatic progression of a developed tumour[5,27,28]. For example, in the well-studied case of breast cancer progression, matrix stiffening by excessive collagen cross-linking disrupts the normal acinar structures of mammary epithelium and promotes invasive phenotype with prominent protrusions at the advanced oncogenic stage[26,27]. In contrast, our work enters a previously unexplored turf of how an existing abnormally stiffened microenvironment might dysregulate EDAC at the initial pre-malignant stage of carcinogenesis. While our findings have several major implications for EDAC and cancer prevention, it remains to be explored how generally

this stiffness-sensitivity applies to other EDAC-related mutations, including those in SRC kinase[51] and transcriptional coactivator YAP[52,53]. Also, given that our study is limited to secondary epithelial cell lines, it remains to be seen how the proposed mechanism holds in primary cell lines. Finally, it will be interesting to ask whether this stiffness sensitivity itself might be tissue-specific in vivo.

Further, related to the origin of stiffened ECM, we note that fibrosis, hyperactive wound healing, chronic inflammation, ageing, and obesity have one common physiological feature —the unusual stiffening of tissue matrix, although the root cause of stiffening could be different in each case[30,31]. Interestingly, all of these pathological conditions also correlate with an elevated risk of cancer. Given that the EDAC mechanism for eliminating the mutation-harbouring cells fails on pathologically stiffened ECM, it is tempting to speculate that these pre-malignant cells are likely to stay in the tissue, acquire more mutations, and ultimately develop into an aggressive tumour over time. In fact, in a mouse model, high fat diet-induced obesity suppresses EDAC-mediated apical elimination of HRas$^{V12}$-transformed cells from intestinal and pancreatic epithelia[54]. Researchers found that this suppression involved both lipid metabolism and chronic inflammation. While they speculated that soluble factors secreted from immune and fibroblast cells might be affecting the competitive interaction between normal and transformed epithelial cells, the underlying molecular mechanism remains unknown[54]. It is thus possible that inflammation-induced ECM stiffness is a key player here, especially considering that increased production of reactive oxygen species (ROS) during inflammation can cross-link and stiffen the ECM.

Interestingly, the fibrotic epithelium is heterogeneous, where stiff and soft ECM patches may be arranged in a mosaic manner. To explore how such heterogeneity might affect EDAC, we created a hybrid ECM gel system where a 4 kPa gel interfaces directly with a 90 kPa gel as a contiguous gel surface (Supplementary Fig. 6a, b, Supplementary Video 5). Under such a scenario, HRas$^{V12}$-transformed cells that were located very close to the interface indeed showed noticeable migration from soft to the stiff side of ECM, possibly by Durotaxis. Moreover, once they migrated to the stiffer side, these transformed cells remained in the monolayer during the entire course of 48 h post-induction with doxycycline. This experiment, therefore, indicated that HRas$^{V12}$-transformed cells experiencing heterogeneous ECM stiffness could evade EDAC and relocate to a more favourable physical microenvironment.

An impending question on the molecular side could be to understand what lies further upstream of Cdc42–filamin interaction on soft ECM and perinuclear cytoskeleton–filamin interaction on stiff ECM. It is well-known that integrin activation by ECM stiffness regulates a variety of downstream effectors, including focal adhesion kinase (FAK) and Rho-GTPases, that aid in actin polymerization and cell contractility[55]. Hence, it is possible that differential Cdc42 activation on soft versus stiff ECM could be a result of the differential integrin activation. In addition, the modalities of stiffness-mediated intracellular protein localization are of great interest in the field, and our work here provides a direction towards the same. Since it is known that the transcriptional coactivator YAP/TAZ acts as a mechanosensor for ECM stiffness[44], its differential localization within the cell can give us a clue about the stiffness-sensitive differential localization of filamin. Given that cytoplasmic localization of YAP/TAZ activates Cdc42 on soft ECM[56], it possibly leads to an enhanced Cdc42 activity at the cell–cell interface. This precedes increased interfacial localization of filamin on soft ECM. On the other hand, increasing evidence also points at the coupling of nucleo-cytoskeletal responses that regulate force transmission from the

matrix to the nucleus mediated via the LINC complex[43]. We can therefore speculate that stiff ECM activates a nuclear-mechanotransduction path involving the LINC complex and FAM101B that regulate perinuclear localization of filamin, which has been elucidated in detail in our work. It is additionally relevant to note a previous study that has shown that application of external force assembles perinuclear actin in a unique Ca$^{2+}$- and INF2 formin-dependent manner[57]. Interestingly, a recent study has found that calcium waves explosively propagating from extruding transformed cells into surrounding normal cells leads to actin reorganization in an INF2-dependent pathway[32]. Considering these results, it will be interesting in future to see how calcium wave interacts with the filamin-dependent pathway during EDAC. Given that Cdc42 and formin activity together decide the ultimate cytoskeletal architecture of a cell, we can guess that Cdc42–filamin axis should have some interesting intersections with Ca$^{2+}$–INF2 axis with respect to EDAC.

Finally, going back to the original question of what micro-environmental factors might decide the success or failure of EDAC, we propose that the molecular mechanism that we have elucidated here offers therapeutic targets to prevent that. Several molecules capable of softening the tissue matrix are already in clinical trials for arresting metastatic progression[27]. We now propose that these molecules can perhaps be used for cancer prevention as well. In addition, disruption of force transmission at various levels, from cell–matrix adhesions to LINC complex, may offer other therapeutic schemes to reduce the risk of cancer. Altogether, this study opens up a new possibility of applying mechanomedicinal strategies to cancer prevention and intends to tip the balance in favour of a successful EDAC.

## Methods

**Cell culture.** Madin–Darby canine kidney (MDCK), Eph4 and Caco-2 epithelial cell lines were used in this study. Tetracycline-resistant wild-type MDCK (MDCK-WT) and HRas$^{V12}$-expressing MDCK (MDCK-GFP-HRas$^{V12}$) cell lines were a gift from Yasuyuki Fujita and were generated as described previously[9]. MDCK cells were cultured in Dulbecco's modified Eagle's medium supplemented with Gluta-Max (Gibco) with 5% foetal bovine serum (tetracycline-free FBS, Takara Bio) and 10 U ml$^{-1}$ penicillin and 10 µg ml$^{-1}$ streptomycin (Pen-Strep, Invitrogen) in an incubator maintained at 37 °C and 5% CO$_2$, unless mentioned otherwise. For setting up cell competition in monolayer, a mosaic monolayer constituting normal MDCK cells (MDCK-WT) and transformed cells (MDCK-GFP-HRas$^{V12}$) were cultured overnight in a specific ratio (40:1) on collagen-coated polyacrylamide (PAA) gels of varying stiffness in the absence of tetracycline (Supplementary Fig. 1a). Cell competition was induced after the monolayer was confluent. Specifically, GFP-HRas$^{V12}$ expression was induced by adding 5 µg ml$^{-1}$ doxycycline (a tetracycline-derived) to the medium. For the control experiments, CellTracker CMAC-Blue (Invitrogen) was used to visualize uninduced MDCK-GFP-HRas$^{V12}$ cells. The cells were first incubated with 10 µM of the dye for 30 min before being mixed with MDCK-WT in the specified ratio (40:1::MDCK-WT:MDCK-GFP-HRas$^{V12}$) and grown overnight on soft substrates (1.2, 4 and 14 kPa). Eph4-Ev cells (ATCC; CRL-3063) were cultured in Dulbecco's modified Eagle's medium supplemented with GlutaMax (Gibco) with 10% foetal bovine serum (tetracycline-free FBS, Takara Bio) and 1.2 µg ml$^{-1}$ puromycin (Gibco) in an incubator maintained at 37 °C and 5% CO$_2$. Caco-2 cells (ATCC; HTB-37) were cultured in Dulbecco's modified Eagle's medium supplemented with GlutaMax (Gibco) with 5% foetal bovine serum (tetracycline-free FBS, Takara Bio) and 10 U ml$^{-1}$ penicillin and 10 µg ml$^{-1}$ streptomycin (Pen-Strep, Invitrogen) in an incubator maintained at 37 °C and 5% CO$_2$. Mosaic monolayers for cell competition using Eph4 or Caco-2 cells were obtained by transfecting either cell monolayers with GFP-HRas$^{V12}$ plasmid and conducting subsequent studies.

To establish stable cell lines expressing mutant proteins for rescuing EDAC on stiff ECM (Fig. 5), MDCK cells were transfected with respective plasmid DNA using Lipofectamine 2000 (Invitrogen). Selection pressure was provided by medium (DMEM-GlutaMAX) containing 400 µg ml$^{-1}$ geneticin (Invitrogen). Stably expressing fluorescent clones were either picked using cloning cylinders (Sigma) following fluorescence confirmation or obtained from single cells. For stable cell line generation from single cells, the transfected cells were initially FACS sorted to obtain a near-homogenous expression of fluorescence. From this fluorescent suspension, single cells were seeded via serial dilution in a 96-well plate and their growth was monitored over two weeks. Post this, homogenously fluorescent colonies derived from single cells were scanned under a fluorescence microscope and sub-cultured into stable cell lines. Subsequent maintenance and

passaging of stable cell lines were done in a medium containing 100 μg ml⁻¹ geneticin. Transient transfection with plasmids was done using Lipofectamine 2000 (Invitrogen) following the manufacturer's protocol. Post 8–12 h of transfection, cells were trypsinized and seeded onto hydrogel substrates and cultured overnight. Upon confluent monolayer generation, cells were either fixed and immuno-stained or imaged directly.

**Hydrogel preparation for compliant ECM.** To provide the cells with the compliant ECM having different stiffness, polyacrylamide hydrogels coated with collagen-I were generated and characterized as described previously[33,58]. 4% (3-Aminopropyl)triethoxysilane (APTES)-treated and 2% glutaraldehyde-activated glass-bottom dishes (Ibidi) were used to cast thin polyacrylamide (PAA) hydrogel substrates. Hydrogel substrates of varying stiffness with an elastic modulus of 1.2, 4, 11, 23, 35, and 90 kPa were prepared by mixing the desired volume of 40% acrylamide and 2% bisacrylamide as given in Supplementary Table 1. Gel surfaces were functionalized with sulfosuccinimidyl-6-(4′-azido-2′-nitrophenylamino) hexanoate (Sulfo-SANPAH, Thermo Scientific) and covalently coated overnight at 4 °C to ensure cell attachment with the following based on experimental requirement: 300 μg ml⁻¹ collagen-I (Invitrogen), 300 μg ml⁻¹ laminin (Merck) or 100 μl BME (Sigma, E6909). Cells were seeded onto the gel area and grown until a confluent monolayer was obtained. Cell competition studies were then carried out.

**Antibodies and plasmids.** Source and dilution information for all primary and secondary antibodies used in immunofluorescence staining are given in Supplementary Table 2. Details of plasmids used in this study are listed in Supplementary Table 3 with their source.

**Immunofluorescence.** Cells were first fixed with 4% formaldehyde diluted in 1× phosphate-buffered saline (PBS, pH 7.4) at room temperature (RT) for 15 min. Following this, cells were washed thrice with 1× PBS. To permeabilize the cells, they were treated with 0.25% (v/v) Triton X-100 (Sigma) in PBS for 10 min at RT followed by washing thrice with PBS to remove the detergent. Non-specific antibody binding was blocked by incubating the samples with 2% BSA in PBST (0.1% v/v Triton X-100 in 1× PBS) at RT for 45 min. Post this incubation time, the blocking buffer was replaced with the primary antibody diluted in blocking buffer and samples incubated at RT for 60 min or at 4 °C overnight. After this, samples were washed twice with PBST and thrice with PBS. Samples were then incubated with secondary antibody tagged with fluorescent dye Alexa Fluor 594 or 647 (Invitrogen) in similar dilution as primary antibody, for 60 min at RT. Counterstaining cell nuclei with a DNA-binding dye 4′,6-diamidino-2-phenylindole (DAPI, 1 μg ml⁻¹ in PBS, Invitrogen) and F-actin with Alexa Fluor dye conjugated phalloidin (Invitrogen) was also done at this step. Finally, samples were washed thoroughly with PBST and PBS before being imaged using confocal microscopy.

**Ultrastructure expansion microscopy.** The ultrastructure expansion microscopy (U-ExM) for filamin and perinuclear cytoskeleton co-localization studies was done as described previously[39], with an optimized expansion condition that retained both antibodies and fluorescent proteins. Also, in this experiment, we transfected the cells with LifeAct-GFP to visualize actin. Briefly, LifeAct-GFP MDCK cells were cultured to confluency on collagen-coated 90 kPa PAA hydrogel and fixed with U-ExM fixation solution, which is 3% formaldehyde (Invitrogen) with 0.1% glutaraldehyde (Sigma) in 1× PBS. Subsequently, the samples were incubated in post-fix solution, which is 0.7% formaldehyde with 1% acrylamide (Sigma) in 1× PBS, overnight at 37 °C. Then, the samples were stained for endogenous filamin with an anti-filamin antibody. Next, to enhance the retention of bound fluorophores during subsequent steps of expansion, the samples were equilibrated with the gel anchoring moiety Acryloyl-X, SE (6-((acryloyl)amino) hexanoic acid, succinimidyl ester; Thermofisher) at a concentration of 0.1 mg mL⁻¹ in 1X PBS for 3 h at room temperature. Then, the samples were immobilized in 100 μL of U-ExM monomer solution composed of 19% sodium acrylate (Sigma), 10% acrylamide (Sigma), 0.1% N,N′-methylene bisacrylamide (Sigma) in 1× PBS supplemented with 0.5% ammonium persulfate (Sigma) initiator and tetramethyl ethylenediamine (Sigma) accelerator, on Parafilm in a pre-cooled humid chamber. Gelation was proceeded for 5 min on ice, and then at 37 °C in the dark for 1 h. Samples were then transferred into ~2 ml of denaturation buffer, made of 50 mM sodium dodecyl sulphate, 200 mM NaCl, and 50 mM Tris in ultrapure water, and the pH was adjusted to 9. The sample denaturation was allowed to proceed for 1 h at 70 °C. After denaturation, samples were placed in deionized water twice every 30 min and then overnight at room temperature. Expanded samples were then trimmed, mounted on 35 mm glass-bottom dishes (Ibidi) and imaged by confocal microscopy.

**Confocal microscopy.** Immunostained samples were acquired using ×60 water objective (UPLSAPO W, N.A. = 1.2, Olympus) mounted on an Olympus IX83 inverted microscope equipped with a scanning laser confocal head (Olympus FV3000), Olympus FV31-SW (v2.3.1.198). Photoconversion, photobleaching and FRET-based sensor studies were done in the same setup using a live-cell chamber supplied with humidified CO₂.

**Filamin localization dynamics using photoconversion and photobleaching.** Photoconversion studies were done on mosaic populations of MDCK-GFP-HRas^V12 cells co-cultured (on soft or stiff ECM) with MDCK cells that had been transiently transfected with mEos2-FilaminA-N-9. To distinguish mEos2-Filamin-expressing green normal cells from GFP-HRas^V12-expressing green transformed cells, we stained the former with CellTracker Blue CMAC (Thermofisher), according to manufacturer protocol, before creating the mosaic monolayer. An optimally expressing mEos2-Filamin cell was chosen that interfaced with an MDCK-GFP-HRas^V12 cell. Stimulation was done on a point region-of-interest in the mEos2-filamin cell using a 405 nm laser at 2% intensity, looped over for 25 times with a scan speed of 1000 μs/pixel. This was immediately followed by LSM imaging of the green and red channels; 0.3% intensity, 700 V PMT voltage for the green and 4% intensity, 650 V PMT voltage for the red channels and filamin dynamics was subsequently tracked for 180 s.

Photobleaching studies were done using MDCK cells stably expressing mApple-FilaminA, cultured on the soft or stiff substrate. 561 nm laser was used at 5% intensity for bleaching a region-of-interest, iterated or looped over five times with a scan speed of 200 μs/pixel. For LSM imaging, the laser power was attenuated to avoid phototoxicity. Images were collected before, immediately after, and for 60 s following the bleaching.

**Förster resonance energy transfer (FRET)-based molecular tension and Cdc42-activity measurements.** FRET experiments for Nesprin tension sensor (Nesprin-TS)[46] were carried out in the live-cell confocal setup (Olympus FV3000). MDCK cells were first plated in the six-well plate (Tarsons) and transiently transfected with Nesprin-TS full length construct. After 12 h, cells were trypsinized and cultured onto soft (4 kPa) and stiff (90 kPa) substrates overnight. Cells were then rinsed and replaced with fresh medium. Images were taken in three different channels: 1. mTFP1: 445 nm laser; filter: 460–500 nm, 2. FRET: 445 nm laser; filter: 530–630 nm, and 3. mVenus: 514 nm laser; filter: 530–630 nm. The pinhole diameter, laser intensity, and exposure times for donor, acceptor, and FRET channels were always kept constant for subsequent experiments. Each field yielded three 1024 × 1024 pixel images representing the donor, FRET, and acceptor channels. Images were then analysed using custom software written in MATLAB (MathWorks). Corrected FRET intensity was calculated by subtracting background and donor bleed-through (dbt) and acceptor cross-excitation. Here, dbt was inferred by the leak-through of mTFP1 signal into the mVenus detector. Acceptor cross-excitation was negligible. The FRET index was quantified by using the pixel-by-pixel intensity FRET ($I_{FRET}$) and donor ($I_D$) channels as: FRET Index = $cI_{FRET}$/($I_D + I_{FRET}$), where $cI_{FRET}$ is the corrected FRET intensity = $I_{FRET} - (dbt \times I_D)$. Nesprin-headless control was also used to establish the appropriate functioning activity of the Nesprin-TS full-length construct.

Raichu-Cdc42 FRET based-biosensor was used for Cdc42 activity measurement in a similar manner to that of Nesprin-TS. Images were taken in three different channels: 1. ECFP: 445 nm laser; filter: 460–500 nm, 2. FRET: 445 nm laser; filter: 530–630 nm, and 3. EYFP: 514 nm laser; filter: 530–630 nm and subsequent analysis was done as mentioned above.

**Inhibition studies.** ML141, an inhibitor of the Rho family GTPase Cdc42 was obtained from Sigma. The powdered chemical form was dissolved in DMSO to make the stock. Cells cultured on the soft substrate were treated with 40 μM ML141 for different time durations to inhibit Cdc42 activity.

**Image analysis.** To measure extrusion count, fixed samples of mosaic monolayers grown on hydrogels of varying stiffness were stained for cell nucleus with DAPI and imaged using ×20 objective on Leica DMI8 inverted microscope using Leica Application Suite X (LAX, v3.7.0.20979). Rounded-up out-of-plane HRas^V12 cell colonies were manually marked as extruded. The fraction of extruded transformed colonies over a total number of transformed colonies per frame was quantified as extrusion count. Approximately 10 frames were acquired per sample (hydrogel of specific stiffness) for each independent experiment. Quantifications shown in various figures were conducted using data from three independent repeats per experiment.

To determine perinuclear and interfacial filamin, ROIs were traced out manually using the selection brush tool (fixed at 10-pixel width) in FIJI (Supplementary Figs. 3a and 4d). Cell-nuclei frame was synced with the filamin frame and used as a reference for tracing the perinuclear region. Interfacial regions were manually traced at the cell–cell interface. Overlapping region tracing was carefully avoided. Mean intensity values were taken for perinuclear and interfacial regions per cell. The fraction of filamin localization per cell for each region was quantified as the ratio of the intensity of the ROI with total intensity. For quantification purposes, the total intensity was the sum of the intensity of perinuclear ROI and interfacial ROI. Fluorescence images were brightness-adjusted and denoised uniformly throughout the whole image for representation purposes only. Denoising was done using the PureDenoise tool in FIJI[59].

**Statistics and reproducibility.** Statistical analyses were carried out in GraphPad Prism 9 (Version 9.2.0). Statistical significance was calculated by Unpaired t-test with Welch's correction or Mann–Whitney test (two-tailed) as mentioned in the corresponding figure. Exact p-values for p < 0.0001 were quantified in MATLAB

R2019b. Scatter-bar plots were displayed as mean ± s.e.m. In the box-and-whisker plot, the centre line denotes the median, the box displays the interquartile range, whiskers indicate range not including outliers (1.5× interquartile range). $p$-Values > 0.05 were considered to be statistically not significant. No statistical methods were used to set the sample size. Quantification was done using data from at least three independent biological replicates. All the experiments with representative images were repeated at least thrice.

**Reporting summary**. Further information on research design is available in the Nature Research Reporting Summary linked to this article.

## Data availability

The authors declare that all quantifications supporting the findings of this study are available within the article and its Supplementary Information or the Source Data File, provided with this paper. Other data, such as raw and processed microscopy images are available from the corresponding author on request. Source data are provided with this paper.

## Code availability

MATLAB-based codes written to analyse FRET experiments used in the current study are available at https://github.com/shilpa10p/TensionSensorFRET.

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

## Acknowledgements

We thank G.V. Shivashankar for critical discussions and suggestions. We also thank Yasuyuki Fujita for cell lines, plasmids, and important inputs. We thank Sanjay Karki, Farah Taj, and Medhavi Vishwakarma for their technical assistance. T.D. is a DBT/ Wellcome Trust India Alliance intermediate fellow and partner group leader of the Max Planck Society (MPG), Germany. This work is funded by DBT/Wellcome Trust India Alliance (grant no. IA/I/17/1/503095 to T.D.). We also acknowledge intramural funds at TIFR Hyderabad from the Department of Atomic Energy (DAE), Govt. of India towards supporting this research and for salaries/fellowships of the authors.

## Author contributions

S.P.P. and T.D. conceived the project. S.P.P performed the majority of the experiments. P.G. exclusively performed expansion microscopy and hybrid PAA gel imaging experiments. P.G. and S.M. contributed to other experiments. S.P.P., P.G., and T.D. analysed the results. S.P.P., P.G., and T.D. wrote the manuscript. All authors agreed on the manuscript as in the submitted version.

## Competing interests

The authors declare no competing interests.

## Additional information

**Peer review information** *Nature Communications* thanks Eduardo Moreno and the other anonymous reviewer(s) for their contribution to the peer review this work. Peer reviewer reports are available.

