## [Peer Review File · Nature Communications]

Reviewers' Comments:

Reviewer #1:

Remarks to the Author:

In this study, the authors investigate which factors influence the extrusion of transformed cells by cell competition with the surrounding normal cells. First, they have found that stiff extracellular matrix (ECM) inhibits the epithelial defense against cancer (EDAC)-associated elimination of transformed cells. They then show several data suggesting that the stiffness of ECM influences Filamin localization that regulates EDAC efficacy; on soft ECM the elevated CDC42 activity induces Filamin localization at cell-cell adhesion sites, while on stiff ECM Filamin interacts with FAM101B and the LINC complex and localizes at the perinuclear region. This study presents conceptually novel and intriguing findings and thus potentially suitable for Nature Communications. However, experimental design of some experiments is not appropriate, and conclusions are not sufficiently supported by the presented data. To consolidate their conclusion or hypothesis, the following concerns need to be addressed before publication.

1. Regarding Figure 4, I am not convinced by the interpretation by the authors (page 6, lines 22-30). It is not clearly described in Material and Method how the expression of those mutants is induced in these experiments. I suppose that according to these heterogenous expression patterns the authors have used the transient transfection method. In MDCK cells, transfection efficiency is rather low, so after transient transfection, expression of exogenous proteins can be observed only in a certain fraction of transfected cells as shown in Fig. 4b-e. However, no obvious phenotypic difference is observed for Filamin localization between mutant-expressing and -non-expressing cells especially in Fig. 4d and e, which cast doubt against the authors' conclusion. The authors should quantify the phenotypic difference between mutant-expressing and -non-expressing cells. Otherwise, the authors should instead use other experimental conditions such as stably expressing clones that are used in Figure 5.

2. In Figure 5, the authors have used only one stably expressing clones for each mutant for analysis. To avoid the concern for clonal effect in selected clones, the authors should test more than two stable clones for each mutant. In addition, the expression level of exogenous mutant proteins and endogenous wild-type proteins should be shown by western blotting to demonstrate that the expression level of mutant proteins is high enough relative to that of wild-type proteins to induce the dominant-negative effects. Furthermore, the homogenous expression of mutant proteins should be demonstrated by immunofluorescence. In page 6, lines 33-35, I do not see the different effect between those mutants as the authors indicate.

3. In Figure 3, the authors focus on cell-autonomous changes in Filamin localization between soft and stiff ECM and do not analyze the non-cell-autonomous changes; according to the previous study (Nature Communications, 2014), Filamin is accumulated at the interface between normal and transformed cells. The authors should check whether CDC42 activity or Tuba membrane localization is further promoted at the interface between normal and transformed cells where Filamin is accumulated.

4. In this study, the authors have used only MDCK cells. To confirm the prevalence of the findings, they should at least check whether the matrix stiffness affects the extrusion of transformed cells using other type(s) of cell lines.

5. In the 3D culture condition, actin is accumulated in the villus at the apical lumen. In Figure 1e, polarized apical localization of actin is seen only in two out of six panels, thus MDCK cells seem not properly polarized in those experiments. Indeed, RasV12-expressing cells look basally delaminated, rather than apically extruded in soft ECM.

Reviewer #2:

Remarks to the Author:

In this MS the authors study the factors involved in tumor-suppressive cell competition, where cells containing oncogenic mutations, like Rasv12, are eliminated by neighboring wt cells. In

particular they study how matrix stiffening prevents tumor-suppressive cell competition. Using photoconversion labelling, protein tracking, and loss-of-function mutations, they attribute this reduced competition to stiffening-induced perinuclear sequestration of a cytoskeletal protein, filamin, that has previously been shown to play a role in tumor-suppressive cell competition. On soft matrix mimicking healthy epithelium, filamin exists as a dynamically single population, which moves to the normal cell-transformed cell interface to initiate transformed cell-extrusion. But, on stiff matrix mimicking fibrotic epithelium, filamin redistributes into two dynamically distinct populations, including a new perinuclear pool, which cannot move to the cell-cell interface. A tug-of-war between filamin-Cdc42 and filamin perinuclear cytoskeleton interactions controls this differential filamin localization and may explain how mechanical stiffening of the extracellular matrix attenuates the elimination of activated HRasV12 cells and the success of tumor-suppressive cell competition on soft versus stiff matrix.

The study is interesting and links observations regarding mechanical forces in cell competition with epithelial defense against cancer (EDAC). It is also well written and results (although in vitro) are convincing. I very much enjoyed the report and only have minor comments and suggestions.

1) The authors classified the 1.2, 4, and 11 kPa ECM substrates as 'soft', mimicking healthy epithelium and substrates with higher levels as 'stiff', potentially mimicking fibrotic epithelium. This classification correlates well with previous studies that reported ECM stiffness of less than 5 kPa in healthy epithelium to 25-100 kPa in fibrotic epithelium. This is interesting however, fibrotic epithelium is normally mosaic, in other words, is likely a patch surrounded by normal or less stiff epithelium. I suggest the authors explore this situation creating patches or regions of stiffer substrate surrounded by soft substrate and see if Rasv12 cells only remain in the stiffer areas and whether there is migration of Rasv12 cells from soft to stiff regions of the cell culture. This is not trivial, because if extruded cells do not die and can migrate to stiffer tissues, EDAC may switch from a tumor suppressive mechanism to a potentially metastasis promoting event.

2) How does the stiffness of the matrix control the subcellular distribution of filamin via the tug of war FAM101B vs. Cdc42? In other words, how exactly does stiffness trigger that signaling event? It is not clear to me how the cells sense stiffness.

3) Does perinuclear filamin play a role, independent of EDAC?

Reviewer #3:

Remarks to the Author:

The submitted work by Pothapragada and colleagues investigates the influence of matrix stiffness on epithelial cell extrusion. Using MDCK monolayers cultured on type I collagen functionalized polyacrylamide hydrogels of varying Young's modulus to model changes in peritumoral ECM stiffness, they find that stiff substrates leads to failed extrusion of induced HRAS-V12 transformed cells. To explore this phenomenon, they then use pharmacologic, mutant protein transfection, and photoconversion experiments to implicate stiffness-induced filamin trafficking as the primary driver of diminished extrusion on stiffer hydrogel surfaces. They propose a stiffness-induced switch based on filamin trafficking between intercellular domains vs perinuclear domains: in cells cultured on soft substrates, filamin localizes at the intercellular interface in a Cdc42 dependent manner, presumably aiding in lamellar extension to enable extrusion; in contrast, in cells on stiff substrates, filamin is recruited through FAM101B to the actin networks that assemble around the nucleus and anchor to the nuclear membrane via the LINC complex. Stiffness induced assembly of LINC complex mediated connections between the cytoskeleton and nucleoskeleton proves to be required for perinuclear filamin localization and impaired extrusion, leading the authors to suggest the idea "mechanomedical" strategies to restore epithelial defense against cancer (EDAC). This work provides new insights into a key mechanism of EDAC and the involvement of ECM stiffness in cytoskeletal and nuclear mechanoregulation. This reviewer believes the manuscript could be suitable for Nature Communications should the follow major and minor critiques be sufficiently addressed.

Major comments:

1. The authors extrapolate their experiments conducted with MDCK monolayers on flat 2D hydrogels to human cancer carcinogenesis, which seems like a leap. While they do include

experiments in a cyst- or acinus-like model, these experiments are poorly described, observational without quantification and statistical analysis, and confusing due to the extrusion of cells basally instead of into the lumen. Because this is an overlay model (2.5D) where the cysts sit atop a gel, one would anticipate the extrusion events or extrusion failure occurs at the interface between the cyst and gel, however the images look like a confocal section at the middle of the cyst above the hydrogel plane. Quantification of frequency as in 2D studies is needed as well as some characterization of location of these events should be included. Additionally, it appears that the formation of a cyst (with open lumen) is inconsistent on stiff gels (Figure 2D) - please comment on the reproducibility of acini formation in the text or provide additional data. Finally, the authors should also add methodological detail for these studies, as the TrueGel3D platform, which has multiple possible compositions (PVA, Dextran, PEG, RGD), is not sufficiently described. This is particularly important for replication of the studies presented in the manuscript as well as for interpretation of their data. For example, are these hydrogels degradable (and via hydrolysis or proteases)? What is the ligand concentrations used?

2. Mechanistic validation of their proposed model in the 2.5D or a true 3D platform would add additional support helping them extend their findings towards in vivo relevance. If there are technical reason why authors cannot conduct further studies in 3D they should consider adding their reasoning in the discussion section.

3. The authors use type I collagen functionalized polyacrylamide gels. This is not a native substrate for epithelial cells adhering to a basement membrane. Can they show that their key findings regarding stiffness hold up with a different functionalization, either functionalizing with laminin/collagen IV or matrigel?

4. Related to the first point, the authors make claims about human cancer prevention and relevance to murine models of cancer but exclusively use MDCK cells for their experiments. While model cell-lines are useful, if results are to be extrapolated to human treatment (on pg. 8 line 43 the authors go as far as to say that stiffness inhibitors should be considered for use in human cancer prevention due to their results) these findings should be validated with human cells, preferably but not limited to a primary cell type. Given numerous studies showing species-specific variability, the robustness of the proposed mechanism should be validated in additional cell types if the work is to be broadly impactful.

5. The duration of the experiments shown is also limited to 24 hours. To broaden their results from basic understanding in a model cell line to human disease they should consider longer-term experiments looking at downstream consequences of hindered extrusion. For example, whether malignant phenotypes begin to form (eg. aberrant proliferation or diminished response to apoptosis) would better connect the results presented in this work to pathology and increase interest from the Nature Communications readership. Similarly, exploring whether cysts become invasive (Figure 1E) would also be informative and increase excitement for the work.

6. Does DN-filamin on in MDCKs on soft matrices lower extrusion efficiency?

7. The "tug-of-war" analogy does not add anything conceptually to the paper and should be removed. As in many other biologies, this is just a situation where a protein localizes distinctly as a function of matrix stiffness due to cytoskeletal reorganization. A "tug of war" would imply that filamin is pulled towards one domain or the other by mechanical linkages to those domains, which is not the author's proposed mechanism. A similar critique can be made of the thermodynamic sink analogy and associated diagrams, which this reviewer suggests should be removed.

Minor comments:

1. Please add sufficient figure labels, including the stiffness of gels used (beyond "soft" and "stiff"), a key for colors used within bar graphs, clear labeling for all fluorescence images, etc.

2. Avoid presenting data without statistics, showing an n=1 representative result without quantification or additional examples in the supplement (Figure 1E, Figure 2C-D, Figure 3D,F, etc.) dampens the claims made. Given the difficulty in quantifying dynamic localization data, the current

method of presentation is not disqualifying but should be avoided wherever possible.

3. Throughout all figures, please show images of cell nuclei – this is particularly important given the implications of nuclear localization presented in the paper. Images should also be presented in greyscale if only one channel is present – the magenta chosen is difficult to see, especially when the manuscript is printed.
4. Diagrams/cartoons should only be included where necessary (typically in the final figure to summarize the mechanistic model). They currently appear throughout each figure and take up considerable figure space. These figures appear to be explaining the experiments, but can have a negative effect of biasing the reader towards the author's interpretation of the experiment. Diagrams that should be removed include: top right of each panel in 1A, panel 1B, panel 2G, panel 4G, panel 5A.
5. In Figure 2, please use greyscale for representative images with only one channel as some colors can be difficult to interpret. This should also be changed throughout the manuscript in subsequent figures.
6. For Figure 2C, please include the comparison to soft and intensity line scans to show the difference in localization, ideally across multiple cells per condition. Also, why is the nucleus LifeAct positive?
7. For 2E please include quantification of nuclear/interfacial filamin localization for this experiment as in 2B. Comparison of cells with and without the construct from the same cultures would help support the paper's findings.
8. In 2F, please comment on the rapid increase in fluorescence depicted in the kymographs, should this be interpreted as due to rapid protein flux or infrequent imaging?
9. Please include additional quantifications from multiple cells for the conditions and experiments shown in Figure 3. This ensures that the data is not being cherry-picked. Additionally, the presentation of 3E is difficult to interpret. It would be more intuitive to present intensity along a line starting from the edge of the nucleus directed towards the cytoplasm.
10. The transfection experiments images in Figure 4 suggest variable transfection efficiency – is there a connection between incorporation of the construct and cell phenotype? It appears that lowly transfected cells are being highlighted with arrows but one would expect these cells to have less pronounced changes in phenotype compared to highly transfected cells. If differences are present, additional data quantifying the differences, or normalization procedures should be included.
11. Generally, the manuscript text needs significant revising for grammar, sentence construction, and flow. There are too many examples to enumerate here and an editing service is advised. One common issue is misappropriation of articles such as "the".
12. The discussion should be drastically condensed to meet the format of the journal. Suggestions for removal of text would be the discussion of cancer treatments and pathology not explored in the manuscript and redundancy in re-explaining the major findings of the work described in the results section.
13. In the discussion, discussion for the role formin, Cdc42, and FAM101B in extrusion should be included, beyond their relative necessity and localization. Relevant cell mechanics work studying Cdc42, filamin, formins, etc. should be referenced. As Nature Communications is a broader readership, the authors need to make it clear why this interaction enables extrusion and EDAC.

Response to Review Comments

Reviewer #1 (Remarks to the Author):

In this study, the authors investigate which factors influence the extrusion of transformed cells by cell competition with the surrounding normal cells. First, they have found that stiff extracellular matrix (ECM) inhibits the epithelial defense against cancer (EDAC)-associated elimination of transformed cells. They then show several data suggesting that the stiffness of ECM influences Filamin localization that regulates EDAC efficacy; on soft ECM the elevated CDC42 activity induces Filamin localization at cell-cell adhesion sites, while on stiff ECM Filamin interacts with FAM101B and the LINC complex and localizes at the perinuclear region. This study presents conceptually novel and intriguing findings and thus potentially suitable for Nature Communications. However, experimental design of some experiments is not appropriate, and conclusions are not sufficiently supported by the presented data. To consolidate their conclusion or hypothesis, the following concerns need to be addressed before publication.

Response: We sincerely thank the reviewer for a thorough review of the manuscript and immensely helpful comments towards its improvement. We have attempted to address all concerns and performed several new experiments. For example, we have reproduced the key experiments in two additional cell lines (Eph4, Caco-2) and added three clones for each mutation study (dnFLNA, dnFAM101B, dnNesprin1, and dnLaminB1). Following, we provide a point-by-point reply to each comment and indicate changes that we have incorporated in the revised manuscript.

1. Regarding Figure 4, I am not convinced by the interpretation by the authors (page 6, lines 22-30). It is not clearly described in Material and Method how the expression of those mutants is induced in these experiments. I suppose that according to these heterogenous expression patterns the authors have used the transient transfection method. In MDCK cells, transfection efficiency is rather low, so after transient transfection, expression of exogenous proteins can be observed only in a certain fraction of transfected cells as shown in Fig. 4b-e. However, no obvious phenotypic difference is observed for Filamin localization between mutant-expressing and -non-expressing cells especially in Fig. 4d and e, which cast doubt against the authors' conclusion. The authors should quantify the phenotypic difference between mutant-expressing and -non-expressing cells. Otherwise, the authors should instead use other experimental conditions such as stably expressing clones that are used in Figure 5.

Response: We thank the reviewer for raising this important point, and we apologize for the confusion caused by our previous representations. For the extrusion experiments related to dnFLNA, dnFAM101B, dnNesprin1, and dnLaminB1 (Figure 5), we had indeed used MDCK cells (clone A) that stably expressed one of these mutant proteins. Relevantly, we had used the same clones for quantification of perinuclear and interfacial filamin fractions as well (Figure 4f). In the revised manuscript, we have now provided correct images using the stable cell lines (Figure 4b-e), which, as predicted, show the nearly homogeneous distribution. In addition, as suggested by the reviewer in the next comment, we have now created two more stable cell lines (clones B and C, Supplementary Figs. 5f-i), through transfection, cell sorting, and clonal selection. Please see a detailed protocol in the revised MATERIALS AND METHODS section (Page No. 9, Lines 31-45). Importantly, all of these cell clones, which stably express one of the aforementioned mutants could rescue the extrusion of HRas^{V12}-transformed colonies on stiff ECM (Supplementary Figures 5j-m). Please also see updated RESULTS section, clearly indicating experiments with these stable cell lines.

2. In Figure 5, the authors have used only one stably expressing clones for each mutant for analysis. To avoid the concern for clonal effect in selected clones, the authors should test more than two stable clones for each mutant. In addition, the expression level of exogenous mutant proteins and endogenous wild-type proteins should be shown by western blotting to demonstrate that the expression level of mutant proteins is high enough relative to that of wild-type proteins to induce the dominant-negative effects. Furthermore, the homogenous expression of mutant proteins should be demonstrated by immunofluorescence. In page 6, lines 33-35, I do not see the different effect between those mutants as the authors indicate.

Response: We thank the reviewer for this very helpful suggestion. As mentioned above, in the revised manuscript, we have raised two additional clones other than the existing one for a specific mutant (dnFLNA, dnFAM101B, dnNesprin1, and dnLaminB1) and performed cell extrusion experiments with all clones. Please see the Supplementary Figures 5j-m. As mentioned above, we have also provided immunofluorescence images for the homogenous expression of mutant proteins. Please see the revised Figure 4b-e and Supplementary Figures 5f-i. We also attempted to perform western blot experiments, as suggested by the reviewer. However, western blot experiments could not capture the ratio of mutant protein to endogenous protein, since two distinct antibodies (specific to mutant and endogenous protein) could not be expected to have identical specificity and binding affinity. Nevertheless, given that we now have identical pieces of evidence from three different clones for each mutant, we believe that these new results are convincing.

3. In Figure 3, the authors focus on cell-autonomous changes in Filamin localization between soft and stiff ECM and do not analyze the non-cell-autonomous changes; according to the previous study (Nature Communications, 2014), Filamin is accumulated at the interface between normal and transformed cells. The authors should check whether CDC42 activity or Tuba membrane localization is further promoted at the interface between normal and transformed cells where Filamin is accumulated.

Response: Following this suggestion, we have now imaged the membrane localization of Tuba (as a measure for CDC42 activity) at the interface between normal and HRas^{V12}-transformed cells. Tuba membrane localization is indeed promoted at the interface between normal and HRas^{V12}-transformed cells, where filamin accumulates. Please see Supplementary Figure 4c and Page No. 5, Lines 23-28 for text.

4. In this study, the authors have used only MDCK cells. To confirm the prevalence of the findings, they should at least check whether the matrix stiffness affects the extrusion of transformed cells using other type(s) of cell lines.

Response: We thank the reviewer for this excellent suggestion, which has definitely helped to increase the generality of our main conclusion. In the revised manuscript, we provide results from two other epithelial cell lines – Eph4 and Caco-2. Eph4 is a nontumorigenic cell line derived from spontaneously-immortalized mouse mammary gland epithelium. Caco-2 is of human origin. Although, it was originally derived from a colon carcinoma, it doesn't show a RAS-phenotype and forms a tight epithelial barrier. Due to these characteristics, researchers have used it extensively for studying the interaction between wild-type and RAS oncogene-expressing cells¹⁻³. Importantly, in Eph4 and Caco-2 cells, ECM stiffness showed similar inhibitory effect on the efficacy of EDAC-associated cell extrusion and perinuclear localization of filamin, as we already had noticed in MDCK cells. These results consolidated the generality of our main conclusion. Please see Supplementary Figures 1f and 3g and Page No. 3, Lines 6-12 for text.

5. In the 3D culture condition, actin is accumulated in the villus at the apical lumen. In Figure

1e, polarized apical localization of actin is seen only in two out of six panels, thus MDCK cells seem not properly polarized in those experiments. Indeed, RasV12-expressing cells look basally delaminated, rather than apically extruded in soft ECM.

Response: We thank the reviewer for this critical comment. We would like to emphasize that for non-monolayer experiments, we cultured the wild-type MDCK cells atop a biochemically-defined hydrogel (TrueGel) that promotes overlaid organotypic culture and whose stiffness can be manipulated within a limited range (1-9 kPa). In this organotypic model, cells generated an enclosed epithelial system (Fig. 1e, Supplementary Fig. 1g) with very distinct and continuous apical sides, as marked by the localization of an apical polarity-marker protein, Golgin-97, and accumulated actin (Supplementary Fig. 1h), as expected in the epithelial lining of an organ. We observe that these organotypic epithelia are morphologically distinct from the classical 3D MDCK cysts, which grow embedded within Matrigel. Considering this discrepancy, we have now removed the cyst terminology when referring to this model and instead call them organotypes. To verify that indeed HRas^{V12}-expressing cells extrude apically and that actin accumulation is towards the apical side, we have now checked the positioning of Golgi apparatus (a well-known apical polarity marker) in our MDCK organotype model. Indeed, Golgi localization follows actin accumulation towards the outer side of the organotypes, and HRas^{V12}-expressing cells extrude in this direction. Consistent with this, we have revised the MDCK organotype images throughout the manuscript. Please see Figures 1e, 2d and Supplementary Figures 1h, 3d, 6a-b, 6d-g. Please also see the updated RESULTS section (Page No. 3, Lines 13-24).

References:

1. Wu, S. K. *et al.* Cortical F-actin stabilization generates apical-lateral patterns of junctional contractility that integrate cells into epithelia. *Nat Cell Biol* **16**, 167-178, doi:10.1038/ncb2900 (2014).
2. Ikonomidou, G. *et al.* Interplay between oncogenic K-Ras and wild-type H-Ras in Caco2 cell transformation. *Journal of proteomics* **75**, 5356-5369, doi:10.1016/j.jprot.2012.06.038 (2012).
3. Gastl, B. *et al.* Reduced replication origin licensing selectively kills KRAS-mutant colorectal cancer cells via mitotic catastrophe. *Cell death & disease* **11**, 499, doi:10.1038/s41419-020-2704-9 (2020).

Reviewer #2 (Remarks to the Author):

In this MS the authors study the factors involved in tumor-suppressive cell competition, where cells containing oncogenic mutations, like RasV12, are eliminated by neighboring wt cells. In particular they study how matrix stiffening prevents tumor-suppressive cell competition. Using photoconversion labelling, protein tracking, and loss-of-function mutations, they attribute this reduced competition to stiffening-induced perinuclear sequestration of a cytoskeletal protein, filamin, that has previously been shown to play a role in tumor-suppressive cell competition. On soft matrix mimicking healthy epithelium, filamin exists as a dynamically single population, which moves to the normal cell-transformed cell interface to initiate transformed cell-extrusion. But, on stiff matrix mimicking fibrotic epithelium, filamin redistributes into two dynamically distinct populations, including a new perinuclear pool, which cannot move to the cell-cell interface. A tug-of-war between filamin-Cdc42 and filamin perinuclear cytoskeleton interactions controls this differential filamin localization and may explain how mechanical stiffening of the extracellular matrix attenuates the elimination of activated HRasV12 cells and

the success of tumor-suppressive cell competition on soft versus stiff matrix. The study is interesting and links observations regarding mechanical forces in cell competition with epithelial defense against cancer (EDAC). It is also well written and results (although in vitro) are convincing. I very much enjoyed the report and only have minor comments and suggestions.

Response: We sincerely thank the reviewer for a thorough review of the manuscript and the kind words of appreciation. Following, we provide a point-by-point reply to each comment and indicate changes that we have incorporated in the revised manuscript.

1. The authors classified the 1.2, 4, and 11 kPa ECM substrates as ‘soft’, mimicking healthy epithelium and substrates with higher levels as ‘stiff’, potentially mimicking fibrotic epithelium. This classification correlates well with previous studies that reported ECM stiffness of less than 5 kPa in healthy epithelium to 25-100 kPa in fibrotic epithelium. This is interesting however, fibrotic epithelium is normally mosaic, in other words, is likely a patch surrounded by normal or less stiff epithelium. I suggest the authors explore this situation creating patches or regions of stiffer substrate surrounded by soft substrate and see if Rasv12 cells only remain in the stiffer areas and whether there is migration of Rasv12 cells from soft to stiff regions of the cell culture. This is not trivial, because if extruded cells do not die and can migrate to stiffer tissues, EDAC may switch from a tumor suppressive mechanism to a potentially metastasis promoting event.

Response: We thank the reviewer for this excellent question, which inspired us to design experiments towards understanding how controlled heterogeneity in ECM stiffness would affect the dynamics of HRas^{V12}-transformed cells. To this end, we created a hybrid polyacrylamide (PAA) gel system where a 4 kPa PAA gel interfaces directly with a 90 kPa gel as a contiguous gel surface (Supplementary Figures 7a-b). Under such a scenario, HRas^{V12}-transformed cells that were located very close to the interface, indeed showed noticeable migration from soft to stiff side of ECM, possibly by durotaxis. Moreover, once they migrated to the stiffer side, these HRas^{V12}-transformed cells remained in the monolayer during the entire course of our experiments, which is up to 48 hours post-induction with doxycycline. This experiment, therefore, indicates that HRas^{V12}-transformed cells experiencing heterogeneous ECM stiffness can evade epithelial defence against cancer (EDAC) by relocating to a more favourable physical microenvironment. This experiment also highlights that the mechanical contribution from ECM has more fundamental influences on the initial stages of oncogenesis than previously thought. We have now included a separate supplementary figure (Supplementary figures 7a-b) and discussed the results of this experiment in the revised manuscript (Page No. 8, Lines 13-22).

2. How does the stiffness of the matrix control the subcellular distribution of filamin via the tug of war FAM101B vs. Cdc42? In other words, how exactly does stiffness trigger that signaling event? It is not clear to me how the cells sense stiffness.

Response: It is well-known that integrin activation by ECM stiffness regulates a variety of downstream effectors, including focal adhesion kinase (FAK) and Rho-GTPases, that aid in actin polymerization and cell contractility⁴. Hence, it is possible that differential Cdc42 activation on soft versus stiff ECM could be a result of the differential integrin activation. In addition, the modalities of stiffness-mediated intracellular protein localization are of great interest in the field, and our work here provides a direction towards the same. Since it is known that the transcriptional coactivator YAP/TAZ acts as a mechanosensor for ECM stiffness⁵, its differential localization within the cell can give us a clue about the stiffness-sensitive differential localization of filamin. Given that cytoplasmic localization of YAP/TAZ activates Cdc42 on soft ECM⁶, it possibly leads to an enhanced Cdc42 activity at the cell-cell interface.

This precedes increased interfacial localization of filamin on soft ECM. On the other hand, increasing evidence also points at the coupling of nucleo-cytoskeletal responses that regulate force transmission from the matrix to the nucleus mediated via the LINC complex⁷. We can therefore speculate that stiff ECM activates a nuclear-mechanotransduction path involving the LINC complex and FAM101B that regulate perinuclear localization of filamin, which has been elucidated in detail in our work. We thank the reviewer for pointing out this important question, and we have discussed it in the revised manuscript (Page No. 8, Lines 23-40).

3. Does perinuclear filamin play a role, independent of EDAC?

Response: Previous investigations have shown that perinuclear filamin is majorly involved in organizing nuclear position and shape via actin cap modulation. For example, during oogenesis within *Drosophila* nurse cells, perinuclear filamin acts as a scaffold for nuclear positioning. Nurse cells employ cytoplasmic actin cables and perinuclear actin to position their nucleus and this positioning is mediated by filamin⁸. Filamin has an early perinuclear localization before actin cable formation and crosslinks them to the perinuclear actin cap. In non-epithelial cells, it is associated with refilin and consequently stabilizes the perinuclear actin cap. There is also an evidence of perinuclear filamin contributing towards the organization of perinuclear actin bundles that accompany EMT in cells stimulated by TGF-beta⁹.

References:

- 4 Geiger, B., Spatz, J. P. & Bershadsky, A. D. Environmental sensing through focal adhesions. *Nat Rev Mol Cell Biol* **10**, 21-33, doi:10.1038/nrm2593 (2009).
- 5 Elosegui-Artola, A. *et al.* Force Triggers YAP Nuclear Entry by Regulating Transport across Nuclear Pores. *Cell* **171**, 1397-1410 e1314, doi:10.1016/j.cell.2017.10.008 (2017).
- 6 Sakabe, M. *et al.* YAP/TAZ-CDC42 signaling regulates vascular tip cell migration. *P Natl Acad Sci USA* **114**, 10918-10923, doi:10.1073/pnas.1704030114 (2017).
- 7 Kirby, T. J. & Lammerding, J. Emerging views of the nucleus as a cellular mechanosensor. *Nat Cell Biol* **20**, 373-381, doi:10.1038/s41556-018-0038-y (2018).
- 8 Huelsmann, S., Ylänne, J. & Brown, N. H. Filopodia-like actin cables position nuclei in association with perinuclear actin in *Drosophila* nurse cells. *Dev Cell* **26**, 604-615, doi:10.1016/j.devcel.2013.08.014 (2013).
- 9 Gay, O. *et al.* RefilinB (FAM101B) targets FilaminA to organize perinuclear actin networks and regulates nuclear shape. *P Natl Acad Sci USA* **108**, 11464-11469, doi:10.1073/pnas.1104211108 (2011).

Reviewer #3 (Remarks to the Author):

The submitted work by Pothapragada and colleagues investigates the influence of matrix stiffness on epithelial cell extrusion. Using MDCK monolayers cultured on type I collagen functionalized polyacrylamide hydrogels of varying Young's modulus to model changes in peritumoral ECM stiffness, they find that stiff substrates leads to failed extrusion of induced HRAS-V12 transformed cells. To explore this phenomenon, they then use pharmacologic, mutant protein transfection, and photoconversion experiments to implicate stiffness-induced filamin trafficking as the primary driver of diminished extrusion on stiffer hydrogel surfaces. They propose a stiffness-induced switch based on filamin trafficking between intercellular domains vs perinuclear domains: in cells cultured on soft substrates, filamin localizes at the intercellular interface in a Cdc42 dependent manner, presumably aiding in lamellar extension

to enable extrusion; in contrast, in cells on stiff substrates, filamin is recruited through FAM101B to the actin networks that assemble around the nucleus and anchor to the nuclear membrane via the LINC complex. Stiffness induced assembly of LINC complex mediated connections between the cytoskeleton and nucleoskeleton proves to be required for perinuclear filamin localization and impaired extrusion, leading the authors to suggest the idea “mechanomedical” strategies to restore epithelial defense against cancer (EDAC). This work provides new insights into a key mechanism of EDAC and the involvement of ECM stiffness in cytoskeletal and nuclear mechanoregulation. This reviewer believes the manuscript could be suitable for Nature Communications should the following major and minor critiques be sufficiently addressed.

Response: We sincerely thank the reviewer for a thorough review of the manuscript and immensely helpful comments towards its improvement. We have attempted to address all concerns and performed several new experiments. For example, we have now reproduced some of the key findings in a 2.5D organotypic model, on basement membrane extract (BME) and laminin-coated gels, and in two additional cell lines (Eph4 and Caco-2). In addition, we have either provided a statistical analysis or backed our conclusion with multiple images, wherever that was required. Following, we provide a point-by-point reply to each comment and indicate changes that we have incorporated in the revised manuscript.

Major comments:

1. The authors extrapolate their experiments conducted with MDCK monolayers on flat 2D hydrogels to human cancer carcinogenesis, which seems like a leap. While they do include experiments in a cyst- or acinus-like model, these experiments are poorly described, observational without quantification and statistical analysis, and confusing due to the extrusion of cells basally instead of into the lumen. Because this is an overlay model (2.5D) where the cysts sit atop a gel, one would anticipate the extrusion events or extrusion failure occurs at the interface between the cyst and gel, however the images look like a confocal section at the middle of the cyst above the hydrogel plane. Quantification of frequency as in 2D studies is needed as well as some characterization of location of these events should be included. Additionally, it appears that the formation of a cyst (with open lumen) is inconsistent on stiff gels (Figure 2D) - please comment on the reproducibility of acini formation in the text or provide additional data. Finally, the authors should also add methodological detail for these studies, as the TrueGel3D platform, which has multiple possible compositions (PVA, Dextran, PEG, RGD), is not sufficiently described. This is particularly important for replication of the studies presented in the manuscript as well as for interpretation of their data. For example, are these hydrogels degradable (and via hydrolysis or proteases)? What is the ligand concentrations used?

Response: We thank the reviewer for this critical comment. We would like to emphasize that our non-monolayer organotypic model is indeed a 2.5D overlay culture where MDCK cells undergo morphogenesis atop the TrueGel setup - a biochemically-defined hydrogel that promotes overlaid organotypic culture and whose stiffness can be manipulated within a limited range (1-9 kPa). In this organotypic model, cells generated an enclosed epithelial system (Fig. 1e, Supplementary Fig. 1g) with very distinct and continuous apical sides, as marked by the localization of an apical polarity-marker protein, Golgin-97, and accumulated actin (Supplementary Fig. 1h), as expected in the epithelial lining of an organ. We observe that these organotypic epithelia are morphologically distinct from the typical 3D MDCK cysts, which grow embedded within Matrigel. Considering this discrepancy, we have now removed the cyst terminology when referring to this model and instead call them organotypes. We have also clarified this issue in the text (Page No. 3, Lines 13-24).

To verify that indeed HRas^{V12}-expressing cells extrude apically and that the actin accumulation is towards the apical side, we have checked the positioning of Golgi apparatus-linked protein, Golgin-97 (a well-known apical polarity marker) in our MDCK organotypic model. Indeed, Golgin-97 localization follows actin accumulation towards the outer side of these organotypes, and HRas^{V12}-expressing cells extrude in this direction. Consistent with this, we have revised the MDCK organotypic images throughout the manuscript. Please see Figures 1e,2d and Supplementary Figures 1h, 3d, 6a-b, 6d-g.

Relevantly, although these organotypes sit atop a gel, the extrusion events we score are predominantly at the interface of the gel and the organotypic model. To make this point clearer, we have included a subfigure in the manuscript where we show a 3D rendered image of the organotypic model in the process of competition (Supplementary Figure 1g) and the representative confocal section, which makes the benchmark for all the subsequent images in the manuscript. We have additionally included the extrusion frequencies in this model as well (Supplementary Fig- 6c) and made the organotypic images consistent throughout the manuscript (Figures 1e, 2d and Supplementary Figures 3d, 6a-g). Please see our clarifications on Page No. 3, Lines 13-24. Finally, we have included a detailed protocol and recipe for creating these TrueGels of different stiffness and how we have cultured these MDCK organotypes for our experiments. Please see the revised MATERIALS AND METHODS section (Page No. 9, Line No. 46 - Page No. 10, Line No. 12) and Supplementary Table 2.

2. Mechanistic validation of their proposed model in the 2.5D or a true 3D platform would add additional support helping them extend their findings towards in vivo relevance. If there are technical reasons why authors cannot conduct further studies in 3D they should consider adding their reasoning in the discussion section.

Response: We thank the reviewer for this extremely interesting suggestion. In the revised manuscript, we indeed validated our key results in the organotypic model (2.5D model). To this end, in addition to the wild-type organotypes, we have now generated organotypes with MDCK cells stably expressing either dnFLNA or dnLaminB1 and transfected a fraction of these cells with the tetracycline-inducible GFP-HRas^{V12} construct and induced transformation with doxycycline. We then studied the effect of ECM stiffness on EDAC and filamin localization in these mutant organotypic epithelia (Supplementary Figure 6a-b). On stiff ECM, for wild-type organotypes (Figure 1e) HRas^{V12}-transformed cells continued to be a part of epithelium even at eight-hour post-induction with doxycycline. However, they started extruding at four-hour post-induction from the mutant organotypic epithelia (Supplementary Figure 6a-b). We also noticed diminished perinuclear filamin in the mutant organotypes (Supplementary Figure 6d-g), further supporting our proposed mechanism. Kindly also see the inclusion of text related to these new experiments on Page No. 6, Lines 42-51.

3. The authors use type I collagen functionalized polyacrylamide gels. This is not a native substrate for epithelial cells adhering to a basement membrane. Can they show that their key findings regarding stiffness hold up with a different functionalization, either functionalizing with laminin/collagen IV or matrigel?

Response: Following this interesting suggestion, we have performed cell competition experiments between the normal and HRas^{V12}-transformed cells on basement membrane extract (BME; Sigma)- and laminin-coated soft (4 kPa) and stiff (90 kPa) polyacrylamide gels. While we used collagen I and laminin in pure form, we deliberately used BME as a mixture of ECM proteins, to establish the generality of our findings. As on collagen-coated gels, we observed significantly lower extrusion of HRas^{V12}-transformed cells on stiff gels than on soft gels for both BME and laminin coating (Supplementary Figure 1d). We also observed significant perinuclear localization of filamin on stiff ECM, coated with either BME or laminin

(Supplementary Figure 3f). Hence, taken together, these new experiments established the generality of our findings. Kindly also see the inclusion of texts related to these new experiments on Page No. 2, Lines 44-47.

4. Related to the first point, the authors make claims about human cancer prevention and relevance to murine models of cancer but exclusively use MDCK cells for their experiments. While model cell-lines are useful, if results are to be extrapolated to human treatment (on pg. 8 line 43 the authors go as far as to say that stiffness inhibitors should be considered for use in human cancer prevention due to their results) these findings should be validated with human cells, preferably but not limited to a primary cell type. Given numerous studies showing species-specific variability, the robustness of the proposed mechanism should be validated in additional cell types if the work is to be broadly impactful.

Response: We thank the reviewer for this excellent suggestion, which has definitely helped to increase the generality of our main conclusion. In the revised manuscript, we provide results from two other epithelial cell lines – Eph4 and Caco-2. Eph4 is a nontumorigenic cell line derived from spontaneously-immortalized mouse mammary gland epithelium. Caco-2 is of human origin. Although, it was originally derived from a colon carcinoma, it doesn't show a RAS-phenotype and forms a tight epithelial barrier. Due to these characteristics, researchers have used it extensively for studying the interaction between wild-type and RAS oncogene-expressing cells¹⁻³. Importantly, in Eph4 and Caco-2 cells, ECM stiffness showed similar inhibitory effect on the efficacy of EDAC-associated cell extrusion and perinuclear localization of filamin, as we already had noticed in MDCK cells. These results consolidated the generality of our main conclusion. Please see Supplementary Figures 1f and Page No. 3, Lines 6-12 for text.

5. The duration of the experiments shown is also limited to 24 hours. To broaden their results from basic understanding in a model cell line to human disease they should consider longer-term experiments looking at downstream consequences of hindered extrusion. For example, whether malignant phenotypes begin to form (eg. aberrant proliferation or diminished response to apoptosis) would better connect the results presented in this work to pathology and increase interest from the Nature Communications readership. Similarly, exploring whether cysts become invasive (Figure 1E) would also be informative and increase excitement for the work.

Response: We thank the reviewer for this interesting suggestion. In the previous version of the manuscript, we had already reported that hindered extrusion on stiff ECM led to long basal protrusions and prominent basal actin fibres (Figure 1d), which were absent in the normal cells. However, following the suggestion, we performed longer-term experiments - up to 60 hours - on stiff ECM where extrusion was hindered. In this case, HRas^{V12}-transformed cells remained in the monolayer, started dividing, and the colony size expanded. It becomes evident at a longer time-point that the transformed-cells colony is not evicted even from an overconfluent monolayer. Despite spontaneous normal cell extrusion occurring in the adjacent areas to maintain homeostasis, HRas^{V12}-transformed cells seem to evade this crowding-dependent cell elimination as evidenced in Supplementary Figure 1e. In the revised manuscript, we have now included one representative longer-term time-lapse imaging where we imaged the monolayer for up to 60 hours (Supplementary Figure 1c, Supplementary Video 2). Extrapolating our *in vitro* data into the physiological domain, we presume that on a fibrotic epithelium, in absence of competition, transformed cells will continue to divide, expand the clones, and will possibly lead to field cancerization over time.

Regarding the possibility of exploring whether organotypic epithelia can become invasive, while we thank the reviewer for this inspiring suggestion, it is technically not possible to

perform invasion experiments with this experimental setup, especially since our MDCK organotype model is cultured atop TrueGel substrates.

6. Does DN-filamin on in MDCKs on soft matrices lower extrusion efficiency?

Response: We did not find any evidence that dnFilamin lowered extrusion efficiency on soft ECM. dnFilamin was designed to jeopardize the perinuclear localization of endogenous filamin.

7. The “tug-of-war” analogy does not add anything conceptually to the paper and should be removed. As in many other biologies, this is just a situation where a protein localizes distinctly as a function of matrix stiffness due to cytoskeletal reorganization. A “tug of war” would imply that filamin is pulled towards one domain or the other by mechanical linkages to those domains, which is not the author’s proposed mechanism. A similar critique can be made of the thermodynamic sink analogy and associated diagrams, which this reviewer suggests should be removed.

Based on our experience while presenting this work to a broader audience, we originally believed that the “tug-of-war” might carry some value in terms of depicting filamin localization on soft and stiff ECM. Biophysicists, especially those using molecular dynamics simulations, like to view protein-protein interaction and the changes in protein conformation as a molecule exploring the landscape and settling in one of local minimum. In addition, the gradient in energy amounts to forces i.e. steeper the descend, the more stable is the conformation and the stronger is the force driving the molecule towards that conformation. Given that Nature Communications is an interdisciplinary journal, we expected that the “tug-of-war” analogy and the representation might draw the attention of the biophysics community. However, we surely agree with the reviewer that this tug-of-war model may not be the highlight of the work, and in the revised manuscript, we have removed the tug-of-war model from Figures 2 and 4. We have also significantly reduced the text related to this model and removed any potentially misleading terms (e.g. ‘pulls’).

Minor comments:

1. Please add sufficient figure labels, including the stiffness of gels used (beyond “soft” and “stiff), a key for colors used within bar graphs, clear labeling for all fluorescence images, etc.

Response: We thank the reviewer for this suggestion for improvement. In the revised manuscript, we have tried our best to add sufficient figure labels, keys for colours used within bar graphs, clear labels for all fluorescence images. While we have made at least some changes in all figures, please find specific changes in Figures 1,2,3 and 5.

2. Avoid presenting data without statistics, showing an n=1 representative result without quantification or additional examples in the supplement (Figure 1E, Figure 2C-D, Figure 3D,F, etc.) dampens the claims made. Given the difficulty in quantifying dynamic localization data, the current method of presentation is not disqualifying but should be avoided wherever possible.

Response: In the revised manuscript, we have either provided statistics or shown multiple images to increase the validity of our claim. For example, we provided statistics in Figs. 3b and 3e, and provided multiple images for same point: Fig. 3d and Supplementary Fig. 4a (top and bottom panels); Fig. 3f and Supplementary Fig. 4b; Fig. 2c and Supplementary Fig. 3c; Fig. 2d and Supplementary Fig. 3d. We have also included line scans and quantification of perinuclear/interfacial filamin localization from multiple cells in Figure 3e. It can be

appreciated that upon ML141 treatment for 60 mins, filamin localization is observed near the nucleus despite cells being cultured on soft ECM.

3. Throughout all figures, please show images of cell nuclei – this is particularly important given the implications of nuclear localization presented in the paper. Images should also be presented in greyscale if only one channel is present – the magenta chosen is difficult to see, especially when the manuscript is printed.

Response: In the revised manuscript, we have provided nuclear images. Please see Figures 1d, 2a, 3c,d,f and 4b-e. Also, we have provided greyscale images for single channels in Figures 2c, 2d, 3c,d,f, 4b-e.

Regarding the images with multiple colours, we would like to point out that the choice of colour combination – cyan, green, magenta – was not arbitrary. In fact, according to the literature, this combination is more accessible to the readers with colour-blindness than the conventional blue-green-red combination. To this end, please refer to the following links:

a. Points of view: Color blindness (<https://www.nature.com/articles/nmeth.1618>)

b. How to make scientific figures accessible to readers with color-blindness (<https://www.ascb.org/science-news/how-to-make-scientific-figures-accessible-to-readers-with-color-blindness/>)

c. Data Visualization with Flying Colors (<https://thenode.biologists.com/data-visualization-with-flying-colors/research/>)

4. Diagrams/cartoons should only be included where necessary (typically in the final figure to summarize the mechanistic model). They currently appear throughout each figure and take up considerable figure space. These figures appear to be explaining the experiments, but can have a negative effect of biasing the reader towards the author's interpretation of the experiment. Diagrams that should be removed include: top right of each panel in 1A, panel 1B, panel 2G, panel 4G, panel 5A.

Response: We appreciate the concern here. Following the suggestion, we have removed several schematics from Figures 2 and 4, including the tug-of-war models.

5. In Figure 2, please use greyscale for representative images with only one channel as some colors can be difficult to interpret. This should also be changed throughout the manuscript in subsequent figures.

Response: We appreciate the concern here. In the revised manuscript, we have made necessary changes in Figures 2c.

6. For Figure 2C, please include the comparison to soft and intensity line scans to show the difference in localization, ideally across multiple cells per condition. Also, why is the nucleus LifeAct positive?

Response: We have now included the comparative line scans for endogenous Filamin localization between soft and stiff substrate (Revised Figure 2c). We have also included another comparative example in the supplementary figure (Supplementary Figure 3c). Please note that these line scans do show localization differences of filamin on soft and stiff ECM. Upon repeated experiments with LifeAct cells, we consistently observed positive nuclear staining for LifeAct. We speculate that this might represent an unavoidable artefact of the technique since fluorescence signal post expansion drops drastically and one needs to image at a higher excitation laser and PMT power. One then tends to pick up the background signal.

7. For 2E please include quantification of nuclear/interfacial filamin localization for this

experiment as in 2B. Comparison of cells with and without the construct from the same cultures would help support the paper's findings.

Response: Following this very helpful suggestion, we have now included quantification of nuclear/interfacial filamin localization in the revised manuscript (Supplementary Figure 3e). Filamin localization increases at the interfacial regions in cells overexpressing mApple-Filamin cultured on stiff ECM, when compared to the wild-type mosaic cultures. As is evident, filamin overexpression compensates for the lack of interfacial filamin pool that was observed during reduced extrusion of HRas^{V12} transformed cells by the wild-type ones.

8. In 2F, please comment on the rapid increase in fluorescence depicted in the kymographs, should this be interpreted as due to rapid protein flux or infrequent imaging?

Response: As rightly pointed out by the reviewer, the apparent rapid increase in protein flux originates from the time lag between subsequent frames. We attempted to reduce the time lag but noticed that acquiring images more frequently amounted to a significant loss of fluorescence due to bleaching.

9. Please include additional quantifications from multiple cells for the conditions and experiments shown in Figure 3. This ensures that the data is not being cherry-picked. Additionally, the presentation of 3E is difficult to interpret. It would be more intuitive to present intensity along a line starting from the edge of the nucleus directed towards the cytoplasm.

Response: We can fully appreciate the concern here. Following this critical suggestion, we have now either provided statistics or shown multiple images to increase the validity of our claim. Please see revised Figures 3b, 3e, Supplementary Fig 4a-b. We have further included more line scans presenting intensity along a line starting near the edge of the nucleus and directed towards the cytoplasm in Figure 3e.

10. The transfection experiments images in Figure 4 suggest variable transfection efficiency – is there a connection between incorporation of the construct and cell phenotype? It appears that lowly transfected cells are being highlighted with arrows but one would expect these cells to have less pronounced changes in phenotype compared to highly transfected cells. If differences are present, additional data quantifying the differences, or normalization procedures should be included.

Response: We thank the reviewer for raising this important point, and we apologize for the confusion caused by our previous representations. For the extrusion experiments related to dnFLNA, dnFAM101B, dnNesprin1, and dnLaminB1 (Figure 5), we had indeed used MDCK cells (clone A) that stably expressed one of these mutant proteins. Relevantly, we had used the same clones for quantification of perinuclear and interfacial filamin fractions as well (Figure 4f). In the revised manuscript, we have now provided correct images using the stable cell lines (Figure 4b-e), which, as predicted, show the nearly homogeneous distribution. In addition, as suggested by the reviewer in the next comment, we have now created two more stable cell lines (clones B and C, Supplementary Fig.5f-i), through transfection, cell sorting, and clonal selection. Please see a detailed protocol in the revised MATERIALS AND METHODS section (Page No. 9, Lines 31-45). Importantly, all of these cell clones, which stably express one of the aforementioned mutants could rescue the extrusion of HRas^{V12}-transformed colonies on stiff ECM (Supplementary Figures 5j-m). Please also see updated RESULTS section, clearly indicating experiments with these stable cell lines.

11. Generally, the manuscript text needs significant revising for grammar, sentence construction, and flow. There are too many examples to enumerate here and an editing service is advised. One common issue is misappropriation of articles such as “the”.

Response: We apologize for any unintentional mistakes in the previous manuscript. In the revised manuscript, we have tried to minimize grammatical errors. We also hope that the reviewer will find this comprehensively edited and revised manuscript an enjoyable read.

12. The discussion should be drastically condensed to meet the format of the journal. Suggestions for removal of text would be the discussion of cancer treatments and pathology not explored in the manuscript and redundancy in re-explaining the major findings of the work described in the results section.

Response: We sincerely thank the reviewer for this helpful suggestion. In the revised manuscript, we have attempted to condense the discussion section, by removing text that either reiterated the experimental results or discussed cancer treatments. However, we chose to discuss a few new issues, as pointed by other reviewers.

13. In the discussion, discussion for the role formin, Cdc42, and FAM101B in extrusion should be included, beyond their relative necessity and localization. Relevant cell mechanics work studying Cdc42, filamin, formins, etc. should be referenced. As Nature Communications is a broader readership, the authors need to make it clear why this interaction enables extrusion and EDAC.

Response: We thank the reviewer for this helpful suggestion. In the revised manuscript, we have now included a discussion on the role of formin, Cdc42, and FAM101B in extrusion. Please see Page No. 8, Lines 40-48.

References:

- 1 Wu, S. K. *et al.* Cortical F-actin stabilization generates apical-lateral patterns of junctional contractility that integrate cells into epithelia. *Nat Cell Biol* **16**, 167-178, doi:10.1038/ncb2900 (2014).
- 2 Ikonomou, G. *et al.* Interplay between oncogenic K-Ras and wild-type H-Ras in Caco2 cell transformation. *Journal of proteomics* **75**, 5356-5369, doi:10.1016/j.jprot.2012.06.038 (2012).
- 3 Gastl, B. *et al.* Reduced replication origin licensing selectively kills KRAS-mutant colorectal cancer cells via mitotic catastrophe. *Cell death & disease* **11**, 499, doi:10.1038/s41419-020-2704-9 (2020).

Reviewers' Comments:

Reviewer #1:

Remarks to the Author:

In the revised manuscript, the authors have responded to most of my comments in an appropriate manner. Therefore, I judge that this paper is, in principle, suitable for publication in Nature Communications, though the authors should address the remaining following (minor) concerns before publication.

Page 3, line 49, Figure 2d: The cyan arrowhead does not indicate interfacial filamin, but the apical side. The authors should show the accumulation of filamin at the interface between normal and RasV12 cells.

I suppose that the labelling for Supplementary Figure 6d-e is quite confusing. For example, does Supplementary Figure 6d show control or dnFLNA mutant? If the former is the case, I do not see the perinuclear staining under the stiff ECM condition. If the latter is the case, the labeling should be corrected like Supplementary Figure 6a, b.

Supplementary Fig6c, right; Soft→Stiff

Text page6 lane25 : Figs. 4b→Figs. 4b-e

Text page6 lane27 : Fig. 4b→Figs. 4b-e

Reviewer #2:

Remarks to the Author:

The authors have satisfactorily answered all my questions and performed new experiments as suggested. I support publication in its present form.

Reviewer #3:

Remarks to the Author:

The revised manuscript by Pothapragada and colleagues addressed many of the reviewers' comments and overall is much improved. In particular, the authors added new experiments confirming their findings with additional cell types and other matrix components, removed extraneous diagrams in the figures, reduced many overstatements in the text, and provided additional characterization towards clarifying their findings in a pseudo-3D model. This work is generally of high quality and should be of interest to the Nature Communications readership. Remaining concerns are listed below:

Major comments:

1. A previous major concern was dissimilarities between the two models employed in these studies. In the majority of studies, cells are plated on top of matrix-functionalized hydrogel substrates with tunable stiffness. In the pseudo-3D model that the authors have renamed "organotypes" (rather dated term, but perhaps the authors are trying to suggest organotypic model?), cells are plated on top of TrueGEL matrices, cultured in a 2% BME containing media, and appear to form cyst-like aggregates with which studies are performed. In this latter model, the authors in their rebuttal explain that apical-basal polarity is apparently flipped from what one would expect for such a structure, where the the apical side is outward. The authors support this statement with golgin-97 immunostaining; to this reviewers knowledge, golgin-97 (a golgi associated protein) is not an established marker of apical-basal polarity and a quick literature search does not suggest it is commonly used as such. Taking the flipped polarity as truth, however, then in this model the stiff underlying substrate is directly adjacent to the apical side of cells in the bottom portion of the cyst. This is flipped compared to the 2D studies. Given the claimed role of Cdc42 and lamellar protrusions along with the established literature of matrix stiffness influence cell adhesion to a substrate, how could the mechanism described in 2D extend to the cyst model given that the apical side of the cell is against the matrix.

The authors added additional quantification for the cyst model, but it's still not clear where extrusions are occurring relative to the underlying gel. Even the differences in extrusion between soft and stiff settings are not evident (eg. Fig 1e at 8 hpi, XZ views). Additionally, there appears to be significant size variation in the formed structures which the authors do not quantitate or describe.

Additionally, if 2% BME is incorporated into this model, what is the argument for using a biochemically defined material?

As currently presented, this reviewer believes the cyst model is sufficiently distinct from the majority of studies using the 2D model and insufficiently characterized, resulting in this data overall confusing the reader more than adding support for the described biology or extending the finding to more physiologic settings. This reviewer suspects the authors are trying to connect their finding to cancer, but the presented data does little to achieve this.

One suggestion would be completely removing all cyst model data throughout the manuscript. Another would be additional characterization and clarifications in the text; the authors need to be very clear to the reader that the polarity of this structure is inverse to that of physiologic epithelial structures.

2. Can the authors demonstrate that extrusion occurs with MDCKs in standard matrigel overlay cultures? Understandably, it would be difficult to modulate stiffness, but this demonstration that at baseline this process occurs for these cells would be a good addition. There is ample literature showing that these cells can form acinar structures with correct apical-basal polarity using this culture method.

3. Controls on soft gels with no tet-induction should be included. The goal would be to demonstrate active extrusion targeting HRAS expressing cells.

4. Authors have repeated studies with two additional cancer cell lines, but still not any primary cells to get as close to primary epithelia as possible. The authors need to be very clear in their discussion that this is a limitation of the current study.

5. Soft/stiff hybrid study is interesting, but perhaps distinct enough of a model to not be included in this work. Currently, the authors first mention the results from this study in the discussion section, which is unusual.

6. Remove redundancies in "Differential localization of filamin..." section (page 3).

7. Remove self-imposed questions throughout text (eg. "Is this filamin localization pattern relevant to EDAC?")

Response to Review Comments

Reviewer #1 (Remarks to the Author):

In the revised manuscript, the authors have responded to most of my comments in an appropriate manner. Therefore, I judge that this paper is, in principle, suitable for publication in Nature Communications, though the authors should address the remaining following (minor) concerns before publication.

Response: We sincerely thank the reviewer for his/her review comments. We have now addressed all remaining minor comments.

Page 3, line 49, Figure 2d: The cyan arrowhead does not indicate interfacial filamin, but the apical side. The authors should show the accumulation of filamin at the interface between normal and RasV12 cells.

Response: We apologize for this inadvertent mistake and thank the reviewer for pointing it out. We have now corrected the position of the arrowhead to indicate the accumulation of filamin at the interface between normal and HRas^{V12} cells. Please refer updated Figure 2d.

I suppose that the labelling for Supplementary Figure 6d-e is quite confusing. For example, does Supplementary Figure 6d show control or dnFLNA mutant? If the former is the case, I do not see the perinuclear staining under the stiff ECM condition. If the latter is the case, the labeling should be corrected like Supplementary Figure 6a, b.

Response: We agree with the reviewer and thoroughly apologize for the confusing labelling. The Supplementary Figure 6d does indeed show dnFLNA mutants, at different time-points post induction with doxycycline. The labels for the same have now been updated, along-with that of dnLaminB1 mutants in the revised manuscript (Supplementary Figure 6d-g).

Supplementary Fig6c, right; Soft→Stiff

Response: We have now corrected the labels in Supplementary Figure 6c.

Text page6 lane25 : Figs. 4b→Figs. 4b-e Text page6 lane27 : Fig. 4b→Figs. 4b-e

Response: We thank the reviewer for pointing out the error in figure labels and have now fixed it in the revised manuscript (Page 6 Line 29 and 31).

Reviewer #2 (Remarks to the Author): The authors have satisfactorily answered all my questions and performed new experiments as suggested. I support publication in its present form.

Response: We sincerely thank the reviewer for the positive assessment and for finding our work suitable of publication.

Reviewer #3 (Remarks to the Author): The revised manuscript by Pothapragada and colleagues addressed many of the reviewers' comments and overall is much improved. In particular, the authors added new experiments confirming their findings with additional cell types and other matrix components, removed extraneous diagrams in the figures, reduced many overstatements in the text, and provided additional characterization towards clarifying their findings in a pseudo-3D model. This work is generally of high quality and should be of interest to the Nature Communications readership. Remaining concerns are listed below:

Response: We appreciate the positive remarks of this reviewer on our study. We have now addressed the remaining comments and also attempted to clarify some concerns regarding the

polarity of our 2.5D organotype model. We hope that the reviewer will now find the revised manuscript suitable for publication.

Major comments:

1. A previous major concern was dissimilarities between the two models employed in these studies. In the majority of studies, cells are plated on top of matrix-functionalized hydrogel substrates with tunable stiffness. In the pseudo-3D model that the authors have renamed “organotypes” (rather dated term, but perhaps the authors are trying to suggest organotypic model?), cells are plated on top of TrueGEL matrices, cultured in a 2% BME containing media, and appear to form cyst-like aggregates with which studies are performed. In this latter model, the authors in their rebuttal explain that apical-basal polarity is apparently flipped from what one would expect for such a structure, where the the apical side is outward. The authors support this statement with golgin-97 immunostaining; to this reviewers knowledge, golgin-97 (a golgi associated protein) is not an established marker of apical-basal polarity and a quick literature search does not suggest it is commonly used as such. Taking the flipped polarity as truth, however, then in this model the stiff underlying substrate is directly adjacent to the apical side of cells in the bottom portion of the cyst. This is flipped compared to the 2D studies. Given the claimed role of Cdc42 and lamellar protrusions along with the established literature of matrix stiffness influence cell adhesion to a substrate, how could the mechanism described in 2D extend to the cyst model given that the apical side of the cell is against the matrix.

Response: In our study, apical-basal polarity of organotypes was demonstrated by immunostaining of F- actin and golgi network, both of which have been established in literature to indicate apical polarity of 3d epithelial cysts. Consequently, we have used Golgin-97 (GOLGA1) here as an established Golgi marker¹⁻³. In fact, in an independent experiment, we performed staining of the epithelial monolayer with anti-Golgin-97 antibody (green) and phalloidin (magenta), as shown below. Scale bar in this image is 10 μ m. This staining clearly reveals apical localization of Golgin-97 (with respect to the nucleus) and accumulated apical F-actin (as shown below). It should therefore eliminate any remaining concerns regarding the polarity of these organotypes.

Regarding the concern about the apical side being adjacent to ECM at the bottom of the organotypes, since we are transfecting the cells with GFP-HRas^{V12} plasmid after the organotype has formed, the bottom surface remains inaccessible to transfection and transformation. However, given that – 1. transformed cells from other surface did extrude apically, 2. the system was ECM stiffness-sensitive, 3. the stiffness-dependent filamin localization could also be observed in this model, and 4. Filamin and other mutants could rescue the EDAC in stiff ECM, it can be concluded there exists a phenomenological and mechanistic homology between this non-monolayer self-organized model and the monolayer model.

The authors added additional quantification for the cyst model, but it's still not clear where extrusions are occurring relative to the underlying gel. Even the differences in extrusion between soft and stiff settings are not evident (eg. Fig 1e at 8 hpi, XZ views). Additionally, there appears to be significant size variation in the formed structures which the authors do not quantitate or describe.

Response: As discussed in our response earlier, the extrusion events occur predominantly at the interface of the gel and the organotype, as can be pictured through the 3d rendering shown in Supplementary Figure 1h. Similarly, the comment that the apical side of organotype is closer to the matrix is not correct, as also evidenced by our 3D rendered view of the model (Supplementary Figure 1h). We hope there are no further confusion related to this issue. Regarding the size variability, it is a characteristics of many self-organized models (intestinal organoids for example). However, we did not find any conspicuous dependence of extrusion frequency on the size of these organoids and in spite of the inherent size variability, the difference between extrusion frequency on soft and stiff ECM remains significant in this model. These results, therefore, indicate that size variability may not be a relevant factor here.

Additionally, if 2% BME is incorporated into this model, what is the argument for using a biochemically defined material?

Response: 2% BME (basement membrane extract or matrigel) in liquid medium is standard “supplement” for culturing MDCK cysts⁴. In this diluted concentration, which is generally used as a supplement or for substrate coating, BME cannot form a gel and hence, does not alter the bulk mechanical properties of defined hydrogel.

As currently presented, this reviewer believes the cyst model is sufficiently distinct from the majority of studies using the 2D model and insufficiently characterized, resulting in this data overall confusing the reader more than adding support for the described biology or extending the finding to more physiologic settings. This reviewer suspects the authors are trying to connect their finding to cancer, but the presented data does little to achieve this. One suggestion would be completely removing all cyst model data throughout the manuscript. Another would be additional characterization and clarifications in the text; the authors need to be very clear to the reader that the polarity of this structure is inverse to that of physiologic epithelial structures.

Response: We would respectfully disagree with the reviewer's suggestion of completely removing all organotypic model data from our manuscript as it is important in lending additional insights into the mechanisms presented in our work. The work is obviously connected to cancer, since the central mutation is HRas^{V12}, and the process described is epithelial defence against cancer (EDAC). Since in the organotypic model, we could show that - 1. Transformed cells did extrude apically, 2. Extrusion in this model was ECM stiffness-sensitive, 3. Filamin localization was stiffness-dependent as in the monolayer model, and 4. Finally, filamin and other mutants could rescue the EDAC in stiff ECM, the model is valuable in extending the scope of our work. We believe that we have also described the advantages and limitations of this model sufficiently in the manuscript. Finally, other reviewers found this model useful and important, and we would like to respect their opinions as well. We believe that since it is difficult to find a biochemically-defined hydrogel that promotes organotypic culture and whose stiffness can be manipulated, our model at this point of time is the state-of-the-art. Whether the community will find this model to be acceptable and useful may be left to the test of time.

2. Can the authors demonstrate that extrusion occurs with MDCKs in standard matrigel overlay cultures? Understandably, it would be difficult to modulate stiffness, but this demonstration

that at baseline this process occurs for these cells would be a good addition. There is ample literature showing that these cells can form acinar structures with correct apical-basal polarity using this culture method.

Response: As pointed out by the reviewer, since the stiffness of matrigel cannot be tuned, doing further extrusion studies on matrigel does not seem to be relevant here. In this respect, we have already very clearly mentioned that we had cultured the wild-type MDCK cells atop a biochemically-defined hydrogel (TrueGel) that promotes organotypic culture and whose stiffness can be manipulated within a limited range (1-9 kPa). In this organotypic model, cells generated an enclosed epithelial system (Fig. 1e, Supplementary Fig. 1h) with very distinct and continuous apical sides, marked by the localization of an apical polarity- marker protein, Golgin-97, and accumulated actin (Supplementary Fig. 1i), as expected in the epithelial lining of an organ.

3. Controls on soft gels with no tet-induction should be included. The goal would be to demonstrate active extrusion targeting HRAS expressing cells.

Response: We thank the reviewer for this suggestion. In the revised manuscript, we have now added the control experiment on soft ECM. Here, the uninduced-HRAS^{V12} mutant cells (labelled with CellTracker CMAC Blue for visualizing) showed significantly less extrusion than the induced system, suggesting that the extrusion reported in our manuscript is majorly active. Please see the Supplementary Figure 1e and corresponding text in the manuscript on Page 3, Lines 1-4 and on Page 9, Lines 27-31.

4. Authors have repeated studies with two additional cancer cell lines, but still not any primary cells to get as close to primary epithelia as possible. The authors need to be very clear in their discussion that this is a limitation of the current study.

Response: Following this suggestion, we have now added additional text clarifying the limitation of this study having not employed the use of primary epithelial cells on Page 7 Line 48-49.

5. Soft/stiff hybrid study is interesting, but perhaps distinct enough of a model to not be included in this work. Currently, the authors first mention the results from this study in the discussion section, which is unusual.

Response: We understand that the soft/stiff hybrid study can be considered as a distinct model, however the purpose of including it in this work originates from a suggestion by Reviewer #2. This experiment highlights the importance of heterogeneity in ECM stiffness influencing the dynamics of HRAS^{V12}-transformed cells, and presents a picture of the mosaic nature of the fibrotic epithelium *in vivo*.

6. Remove redundancies in “Differential localization of filamin...” section (page 3).

Response: Following this suggestion, we have now attempted to modify some parts of the text in this section in order to remove redundancies.

7. Remove self-imposed questions throughout text (eg. “Is this filamin localization pattern relevant to EDAC?”)

Response: We have implemented the suggested changes throughout the revised manuscript, which does not have any sentence ending with a question mark.

References:

1. Evans, J.H., Spencer, D.M., Zweifach, A. & Leslie, C.C. Intracellular calcium signals regulating cytosolic phospholipase A2 translocation to internal membranes. *The Journal of biological chemistry* **276**, 30150-30160 (2001).
2. Ducharme, N.A. *et al.* Rab11-FIP2 regulates differentiable steps in transcytosis. *American journal of physiology. Cell physiology* **293**, C1059-1072 (2007).
3. Alzhanova, D. & Hruby, D.E. A trans-Golgi network resident protein, golgin-97, accumulates in viral factories and incorporates into virions during poxvirus infection. *Journal of virology* **80**, 11520-11527 (2006).
4. Bryant, D.M. *et al.* A molecular network for de novo generation of the apical surface and lumen. *Nat Cell Biol* **12**, 1035-1045 (2010).

Reviewers' Comments:

Reviewer #3:

Remarks to the Author:

My major issue is the difference between the two models they present: they use 1) epithelial monolayers to show underlying matrix stiffness drives upwards, apical extrusion of HRAS cells, and in a subset of studies, 2) acini plated on top of a monolayer on top of a hydrogel with tunable stiffness. The latter has flipped apical-basal polarity (apical side is on the outside) to what is normally seen in such structures (even produced with the same cell line) and the basal side of the epithelia is inward not juxtaposed with stiff matrix. Given the marked differences between these two models, the limited rigor to which they show parallel findings between the two settings, and how cagey they are with being honest about the differences between the models, my feeling is this work is misleading and will create confusion or skepticism from most readers. At the minimum, they need to blatantly state that the polarity is reverse to what would be expected for such an acinar structure. I asked for this change in the last revision, but despite the current rebuttal promising so, I do not see anything to this effect in the revised text. Also, they need to be open about the fact that what stiffness (underlying monolayer of cells, matrix juxtaposed with the apical side) the cells are responding to is not clear in the organotypic model and that likely it is distinct from the monolayer studies.

Response to Review Comments

Reviewer #3 (Remarks to the Author):

My major issue is the difference between the two models they present: they use 1) epithelial monolayers to show underlying matrix stiffness drives upwards, apical extrusion of HRAS cells, and in a subset of studies, 2) acini plated on top of a monolayer on top of a hydrogel with tunable stiffness. The latter has flipped apical-basal polarity (apical side is on the outside) to what is normally seen in such structures (even produced with the same cell line) and the basal side of the epithelia is inward not juxtaposed with stiff matrix. Given the marked differences between these two models, the limited rigor to which they show parallel findings between the two settings, and how cagey they are with being honest about the differences between the models, my feeling is this work is misleading and will create confusion or skepticism from most readers. At the minimum, they need to blatantly state that the polarity is reverse to what would be expected for such an acinar structure. I asked for this change in the last revision, but despite the current rebuttal promising so, I do not see anything to this effect in the revised text. Also, they need to be open about the fact that what stiffness (underlying monolayer of cells, matrix juxtaposed with the apical side) the cells are responding to is not clear in the organotypic model and that likely it is distinct from the monolayer studies.

Response: We thank the Reviewer for the insightful review comments and for appreciating our work as *high quality and of interests to Nature Communications during previous rounds*. Most importantly, we thank him/her for taking the trouble to go through multiple versions of our manuscript and evaluate each and every aspect of the manuscript critically. Efforts from this Reviewer towards reviewing our manuscript epitomizes how the critical peer-review process should work in science, and we really appreciate it. These comments have definitely improved the quality of the work and helped us in crystalizing our main discoveries in a concise manner. We also sincerely apologize for any inconvenience caused by the multiple revisions. In past, we have implemented several changes recommend by this Reviewer, including using different ECM-coating, using additional cell lines, performing longer-term experiments, performing control experiments without induction, and removing the ‘tug-of-war’ analogy, to list a few. We would now request him/her to consider **a revised version of the manuscript, where we have now removed the organotypic section entirely from the main manuscript and the supplementary, thus addressing the only remaining major concern of the Reviewer.** This suggestion was in fact provided by this Reviewer in the previous round of review comments (revision 2), as he/she had mentioned “**One suggestion would be completely removing all cyst model data throughout the manuscript**”. In the previous round, we tried our best to validate the issues related to the mechanistic and phenotypic homology between the monolayer and the organotypic model. We had provided several representations and new experiments characterizing the polarity of the latter model and the plane of extrusion in this 2.5D system. There are limitations with respect to performing these experiments in 3D, mostly due to the absence of a hydrogel system that promotes epithelial morphogenesis and whose stiffness can be tuned within a significantly wide range. What we had attempted under these constraints, was perhaps the state-of-the-art. We hoped the Reviewer would agree with our reasoning. However, at this point, we do agree with the Reviewer that there still exist some differences, which may create confusion to some readers. To this end, as also endorsed by the editor, removing the data related to the organotypic model seemed to be the only reasonable course of action. One alternative to this action could have been retaining all results related to the organotypic model in few separate supplementary figures and discuss the findings and limitations of this model in the supplementary text, if and only if the Reviewer agrees/recommends that. However, *given that removing the organotype-results does not*

compromise the quality of the findings, especially since the epithelial monolayer system has been a 'gold standard' in EDAC research¹⁻³, we would be equally happy if the present revised version without the organotypic system is considered for publication. Hence, ***considering that at this point, the only remaining concern against the publication of this manuscript is the homology between the monolayer and organotypic system, we hope that the Reviewer will find the removal of the organotypic section as an appropriate response and recommend the publication of this manuscript in its present format.***

References:

1. Kon, S. *et al.* Cell competition with normal epithelial cells promotes apical extrusion of transformed cells through metabolic changes. *Nat Cell Biol* **19**, 530-541 (2017).
2. Kajita, M. *et al.* Filamin acts as a key regulator in epithelial defence against transformed cells. *Nat Commun* **5** (2014).
3. Wagstaff, L. *et al.* Mechanical cell competition kills cells via induction of lethal p53 levels. *Nat Commun* **7** (2016).

Reviewers' Comments:

Reviewer #3:

Remarks to the Author:

The authors have addressed all of my concerns and I believe this manuscript is suitable for publication in Nature Communications.

Response to Review Comments

Reviewer #3 (Remarks to the Author):

The authors have addressed all of my concerns and I believe this manuscript is suitable for publication in Nature Communications.

Response: We sincerely thank the Reviewer for supporting the publication of our manuscript in Nature Communications and for his/her insightful review comments. Addressing these comments has improved the manuscript quality during the review process.